# Integrating cryo-OrbiSIMS with computational modelling and metadynamics simulations enhances RNA structure prediction at atomic resolution

Shannon Ward [1,2,6], Alex Childs[1,2,6], Ceri Staley[1,6], Christopher Waugh[1,2,3], Julie A. Watts [4], Anna M. Kotowska [4], Rahul Bhosale [5] & Aditi N. Borkar [1,2,3] ✉

The 3D architecture of RNAs governs their molecular interactions, chemical reactions, and biological functions. However, a large number of RNAs and their protein complexes remain poorly understood due to the limitations of conventional structural biology techniques in deciphering their complex structures and dynamic interactions. To address this limitation, we have benchmarked an integrated approach that combines cryogenic OrbiSIMS, a state-of-the-art solid-state mass spectrometry technique, with computational methods for modelling RNA structures at atomic resolution with enhanced precision. Furthermore, using 7SK RNP as a test case, we have successfully determined the full 3D structure of a native RNA in its apo, native and disease-remodelled states, which offers insights into the structural interactions and plasticity of the 7SK complex within these states. Overall, our study establishes cryo-OrbiSIMS as a valuable tool in the field of RNA structural biology as it enables the study of challenging, native RNA systems.

Biomolecular complexes, specifically those involving RNAs and their protein partners, exhibit intricate 3D organisation and intermolecular interactions that govern their cellular functions. Therefore, understanding the architecture and dynamics of these complexes is of great significance in unravelling disease pathways, advancing drug discovery, and designing targeted therapeutics. Although conventional techniques such as X-ray crystallography, Cryo-Electron Microscopy, and NMR spectroscopy have greatly enhanced our knowledge of biological macromolecules[1], they do have certain limitations, especially when it comes to sample preparation and capturing structures in their native state. These techniques require highly pure, stable, and homogeneous samples, yet native RNA complexes are often low in abundance, exist within heterogeneous mixtures, and exhibit flexible and transient interactions. SHAPE-MaP is another widely used experimental technique for determining the secondary structures of a wide range of RNA systems at nucleotide resolution. However, the RT-PCR amplification and sequencing steps in SHAPE-MaP workflows could, in practice, impose a lower limit on the size of the RNAs that can be reliably studied using this technique. Furthermore, SHAPE-MaP and other chemical probing techniques typically require microgram quantities of sample (Supplementary Note 1), which can be challenging to generate for native RNA systems. Therefore, there is a pressing need for integrative approaches capable of overcoming these challenges and investigating such systems at high-resolution[2,3].

The advent of OrbiSIMS[4], a breakthrough in biological solid-state mass spectrometry, holds great promise in addressing this research

[1]School of Veterinary Medicine and Science, University of Nottingham, Nottingham LE12 5RD, UK. [2]Wolfson Centre for Global Virus Research, University of Nottingham, Nottingham LE12 5RD, UK. [3]RHy-X Limited, London WC2A 2JR, UK. [4]School of Pharmacy, University of Nottingham, Nottingham NG7 2RD, UK. [5]School of Biosciences, University of Nottingham, Nottingham LE12 5RD, UK. [6]These authors contributed equally: Shannon Ward, Alex Childs, Ceri Staley. ✉e-mail: aditi.borkar@nottingham.ac.uk

gap. In OrbiSIMS, an incident argon gas cluster ion beam triggers ballistic fragmentation of the specimen while retaining relatively large secondary ions that are subsequently analysed by the Orbitrap mass analyser. This enables in situ chemical analysis of various materials, ranging from hard and soft matter to biological cells, tissues and macromolecules such as proteins, with exceptional mass resolution and picomolar sensitivity[5–11].

In this study, we demonstrate the feasibility of cryogenic OrbiSIMS (cryo-OrbiSIMS) and its integration with molecular modelling and enhanced computer simulations to determine high-resolution 3D structures of RNAs. Through rigorous benchmarking against existing methodologies, we thoroughly evaluated the accuracy, reliability and limitations of the technique. Our findings demonstrate that the chemical information encoded in the cryo-OrbiSIMS spectrum correlates not only to the biochemical composition, but also to the structural characteristics of the native RNA complexes. By incorporating this data as restraints in molecular modelling algorithms and metadynamics simulations, we constructed native-like 2D-folds and atomic resolution 3D-structures, respectively, of the studied RNA systems. Using this benchmarked pipeline, we investigated the complete 3D structure of the 7SK RNA in its apo and protein-bound states, shedding light on the structural changes induced by HIV Tat protein during viral infection.

In summary, our results establish key performance metrics such as sensitivity, resolution, and reproducibility, for cryo-OrbiSIMS as a valuable tool in the field of RNA structural biology. This paradigm shift can enable and accelerate the study of challenging biomolecular structures, opening up avenues for understanding the intricate architecture of these complexes and their functional implications.

## Results

### Cryo-OrbiSIMS fingerprint can differentiate structurally distinct RNAs

OrbiSIMS has been previously benchmarked for characterising proteins in situ and from a chemical mixture[5]. Despite this ability of the method for chemical analysis of macromolecules, its suitability for studying RNA and RNA-protein complexes remained unexplored. To optimise the sample preparation and data collection conditions for such systems, we used the bacterial ribosome as a representative native RNA-protein complex, as it is highly abundant and functionally relevant in the cell. In room temperature experiments (Fig. 1a and Supplementary Fig. 1), we observed mass fragments for the ribosomal RNAs (rRNAs) and proteins up to $m/z = 750$ and achieved a mass accuracy of 2 ppm. However, under cryogenic conditions, we observed a remarkable increase in the mass of detected fragments up to $m/z = 2250$, while retaining the mass accuracy of 2 ppm. Particularly for the rRNAs, for which fragments are detected in the negative polarity spectrum (Fig. 1a), we found common peaks (within a 2 ppm error) between room temperature and cryogenic conditions. These accounted for 5% of the total peaks, covering 12.7% of the intensity area, in the room temperature data and only about 0.48% of the total peaks, covering 3.67% of the intensity area, in the cryogenic OrbiSIMS data. These results are consistent with better ionisation yields[12] in cryo-OrbiSIMS compared to room temperature conditions due to hydrogen donation from $H_3O^+$ in the water matrix and due to molecules retaining their native state[13].

To assess the consistency in generating such cryo-OrbiSIMS data for a given sample, we analysed a total of 38 datasets comprising of nine different types of RNA systems (Supplementary Table 1). These systems represent the Cas9 protein:sg RNA complex and bacterial ribosomes under wildtype, mutant, denaturing and native conditions. Our results indicate that, at a 1% significance level, all the technical repeats for each system are derived from the same mass distribution (Supplementary Figs. 2 and 3). Furthermore, the repeat data had an average spectral overlap of 35.83% and 37.49%, and the mean Wasserstein distance metric was 15.36 and 8.19 for the negative and

positive polarity data, respectively. These findings strongly indicate that cryo-OrbiSIMS measurements are reproducible, making it a reliable method for studying various RNA complexes.

Interestingly, when we compared the nine RNA systems with each other, we found that the average Wasserstein distances between the samples increased to 220.67 and 55.65 for the negative and positive polarity data respectively. At the same time, the spectral overlap decreased to 0.4% and 1.8%, for the negative and positive polarity data, respectively. This finding suggests that cryo-OrbiSIMS can differentiate between various RNA complexes based on their spectral profiles. To investigate this further, we performed z-score clustering analysis on the data, which normalized the Wasserstein distances between the samples to have a mean of 0 and a standard deviation of 1. As expected, the technical replicates of each sample in both polarities consistently grouped together. Furthermore, for the positive polarity data (Supplementary Figs. 1b and 3), we observed a clear distinction between the ribosomal samples and the Cas9:sg samples as the proteins in these samples are distinct. However, in the negative polarity data (Fig. 1b and Supplementary Fig. 2), the denatured ribosome sample clustered alongside the apo sg RNAs, separate from the native ribosome sample. This outcome is noteworthy because the rRNAs in the native and denatured ribosomal samples have identical sequences. Similarly, the apo sg RNAs clustered away from the Cas9 bound forms, even though the RNA components in these samples are identical. In conclusion, the differences we observed in the clustering likely stem from differential cryo-OrbiSIMS fragmentation between the RNAs in their apo forms and their protein bound forms, potentially reflecting differences in their structure and /or native contacts within these states.

### Cryo-OrbiSIMS fragmentation depends upon native RNA contacts

In cryo-OrbiSIMS, an incident $Ar_{3000}^+$ gas cluster ion beam causes ballistic fragmentation of the sample, generating the mass spectrum. Thus, unbound, or exposed regions are expected to experience greater fragmentation, resulting in a higher number of peaks in the cryo-OrbiSIMS spectra. In contrast, regions that are bound or buried within the complexes are anticipated to exhibit fewer peaks[14]. Based on this understanding, we proposed a hypothesis that the number of times a particular residue is assigned within different ionised fragments of the sample, and subsequently its association with multiple peaks in the cryo-OrbiSIMS spectrum, directly corresponds to its exposure or burial within the studied RNA system. Thus, by analysing these assignment frequencies, we may gain insights into the accessibility of the RNA residues within the complex and, by extension, its structure.

To test our hypothesis, we assigned peaks for the sg RNA and 5s rRNA residues in the native Cas9:sg and ribosome samples respectively (Supplementary Information Spreadsheet) and examined their relationship with the respective RNA residue's solvent accessible surface area (SASA) and its number of native contacts in the parent complex. Contrary to our initial hypothesis, we observed a slight negative correlation between the frequency of residue assignments and SASA (Supplementary Fig. 5). In simpler terms, residues less exposed to the solvent tended to have higher assignment frequencies. Moreover, we found a slight positive correlation between the frequency of residue assignments and the number of its native contacts (Supplementary Fig. 5). This suggests that residues with more native contacts, indicating a greater degree of structural rigidity or being buried within the structure, were more frequently assigned in the cryo-OrbiSIMS spectra.

To gain a deeper understanding of the relationships between these parameters, we performed a principal component analysis (PCA) on the data (Fig. 1c, d). The PCA results showed that the number of native contacts and the frequency of residue assignments mainly contributed to the first and second principal components, respectively, and explained more than 95% of the variability in the data. Interestingly, the contributions to the third principal component

displayed distinct patterns between the sg RNA and 5s rRNA systems. In the case of the Cas9:sg system (Fig. 1c), the third principal component was influenced by whether the RNA residues were base paired or unpaired. Conversely, for the 5s rRNA within the ribosome (Fig. 1d), the SASA parameter significantly and positively contributed to the third

principal component. Nonetheless, when we projected the experimental data onto the principal components for both systems, we observed a clear distinction between the average assignment frequencies between unpaired and paired RNA residues (Fig. 1e). Unpaired residues located at the RNA:protein interaction sites and

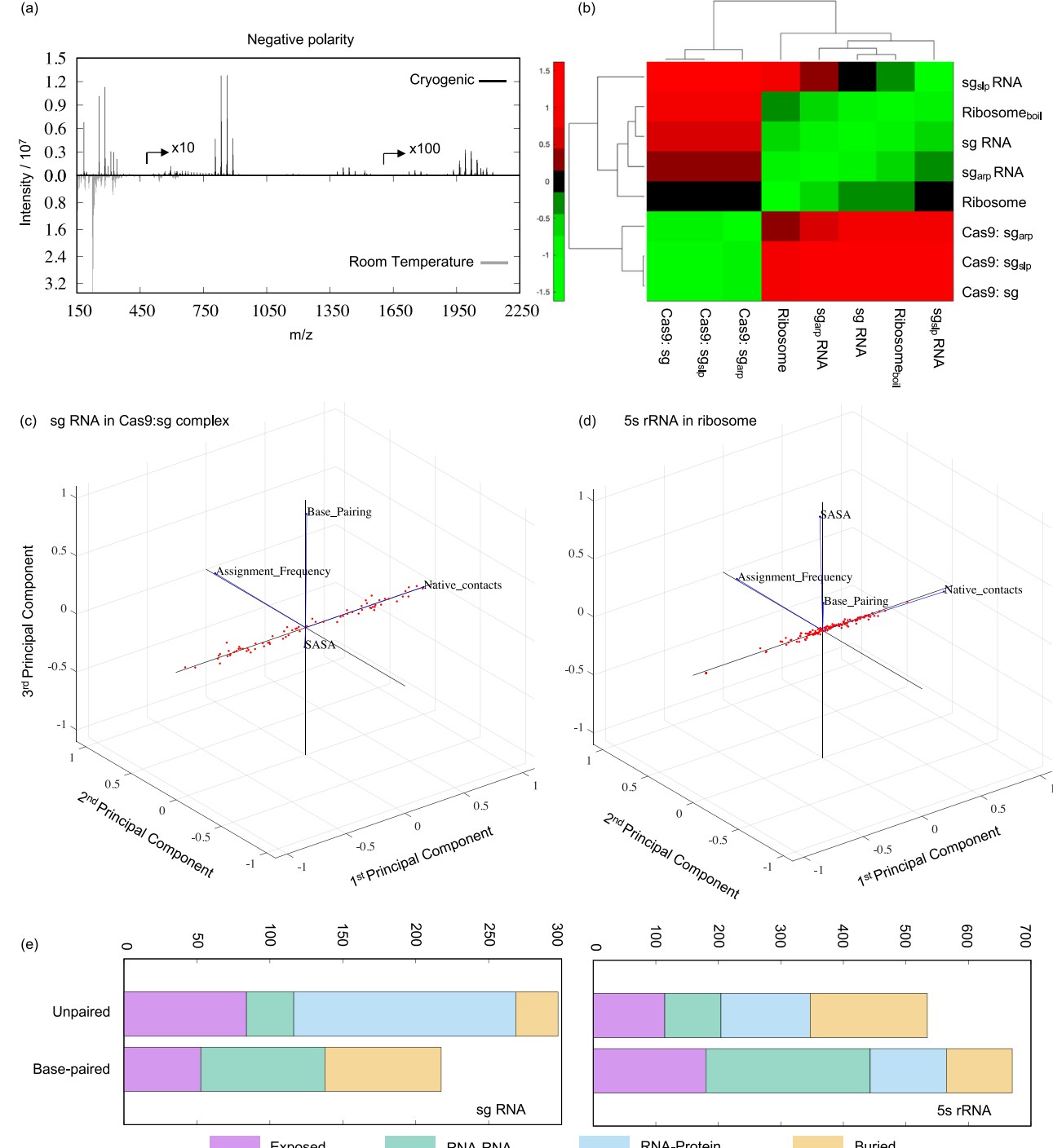

**Fig. 1 | Cryo-OrbiSIMS optimisation for RNA-protein systems. a** Comparison of negative polarity OrbiSIMS spectrum of native bacterial ribosomes under room temperature and cryogenic conditions. Peak intensity for fragments $m/z > 500$ is multiplied by 10, and that for fragments $m/z > 1500$ is multiplied by 100 for better visualization. **b** Hierarchical clustering of the negative polarity cryo-OrbiSIMS data shows that cryo-OrbiSIMS can distinguish biochemically identical but structurally distinct RNAs. **c, d** Principal component analysis of the (**c**) sg RNA and (**d**) 5s rRNA peak assignment frequencies, solvent accessible surface area (SASA) values and

native contacts within their respective RNP complex. Blue vector lines denote the respective parameter, and red dots denote the parameter values projected onto the principal components. For both the RNA systems, their residue assignment frequency and native contacts together account for more than 95% variability in the data and significantly contribute to the 1st and the 2nd principal components, respectively. **e** Cumulative assignment frequencies for the sg RNA and 5s rRNA residues classified according to their base pairing, solvent exposure, burial, or native contact characteristics in the parent RNP complex.

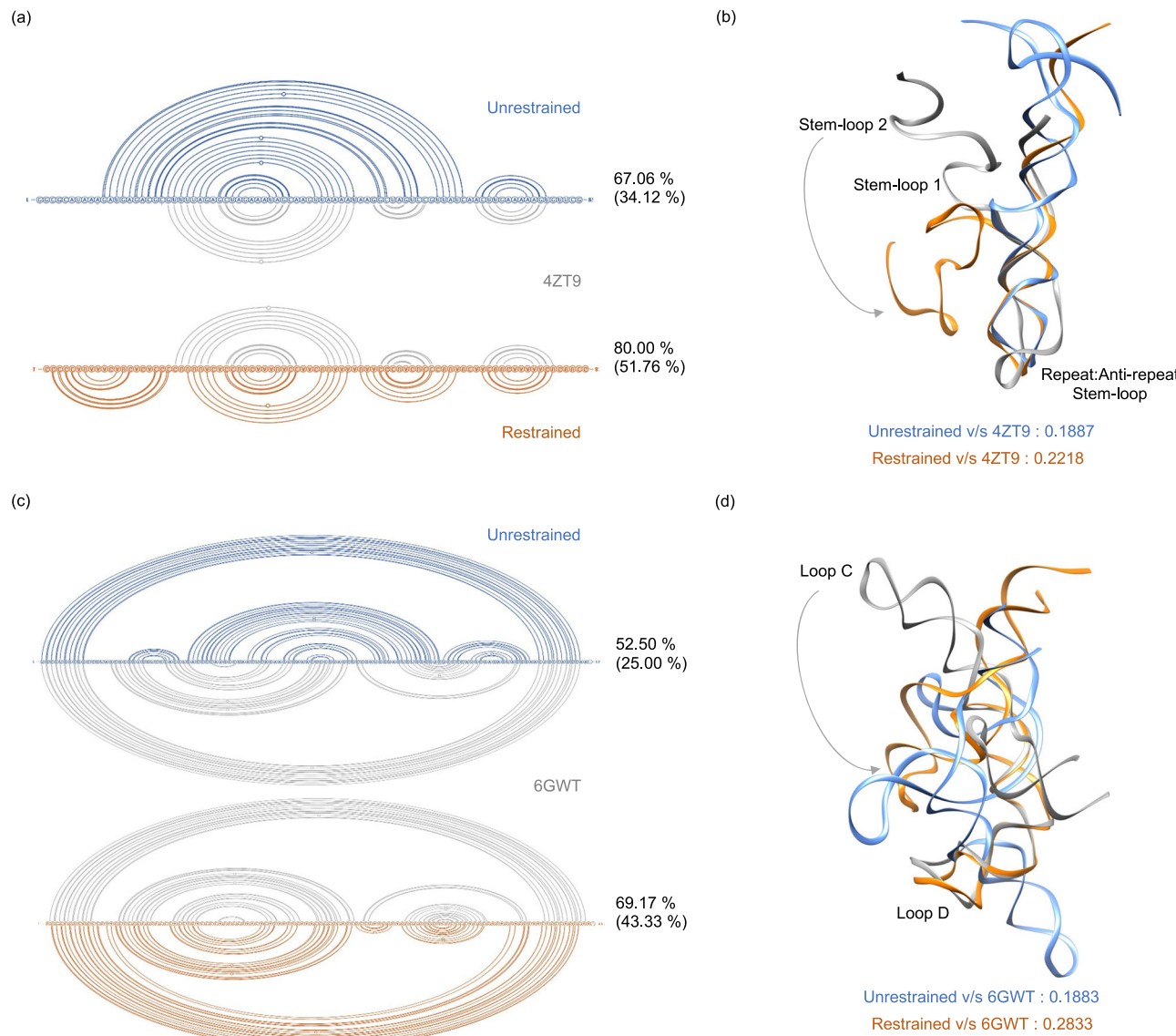

**Fig. 2 | Cryo-OrbiSIMS guided modelling of sg RNA and 5s rRNA structures.** Panels (**a**, **b**) denote sg RNA and panels (**c**, **d**) denote 5s rRNA structures. 2D and 3D structures are denoted via linear and ribbon representation respectively. Unrestrained, cryo-OrbiSIMS restrained, and PDB structures are coloured blue, orange and grey, respectively. The 2D structure predictions are compared using percent structure similarity (and percent structure identity in brackets) calculated using the Beagle tool[18]. The 3D structures are compared using the TM-Score metric calculated using US-align[29]. The modelling shows that, when compared to the unrestrained prediction, the restrained 2D folds are significantly similar to the PDB structure[17,19] of the corresponding RNA in its native complex. The 3D structures predicted using these 2D folds also exhibit the same comparison trend and have significantly lower RMSD values for the structurally aligned residues in restrained structures. However, when comparing the global conformation of these RNAs, stem-loop 2 of the sg RNA (**b**) and loop C in 5s rRNA (**c**) seem to be flipped by 180° in the cryo-OrbiSIMS restrained structure relative to the PDB structure.

base-paired residues at the RNA:RNA interaction sites exhibited the highest assignment frequencies. This suggests that the parent RNA molecule is most likely to undergo fragmentation at these sites within the RNA-protein complex. These findings highlight the impact of native interactions rather than solvent exposure of the RNA residues on the cryo-OrbiSIMS mass data. Thus, we propose a refined working hypothesis that the peak assignment frequency for an RNA residue is related to its number of native contacts in the corresponding RNA-protein complex.

**Cryo-OrbiSIMS-guided modelling of precise RNA 3D structures**
Our findings above have shown that the cryo-OrbiSIMS data contains valuable information about the structural aspects of RNA residues within their native complexes, particularly regarding their native contacts. We utilised this information by using the cryo-OrbiSIMS peak assignment frequency for each residue as base-pairing probabilities and used these probabilities as restraints when predicting[15,16] the secondary structure (2D) folds of sg RNA and the 5s rRNA sequences (Fig. 2, Supplementary Figs. 6, 7). This approach yielded notable improvements in the secondary structure prediction. For sg RNA, the use of restraints increased the percent similarity between the predicted structure and the PDB structure[17] by 1.2-fold[18] and identity by 1.5-fold (Fig. 2a). Similarly, for the 5s rRNA[19], these values increased by 1.3-fold and 1.7-fold, respectively (Fig. 2c). Furthermore, comparing this cryo-OrbiSIMS restrained structure with the previously determined chemical probing-restrained 2D structure of the 5s rRNA[20-23] (Supplementary Table 2) reveals that cryo-OrbiSIMS restraints perform equally well as the chemical mapping techniques for determining RNA 2D structures. We next used the restrained 2D folds to model the tertiary (3D) structures of these RNAs[24,25]. As expected, incorporating

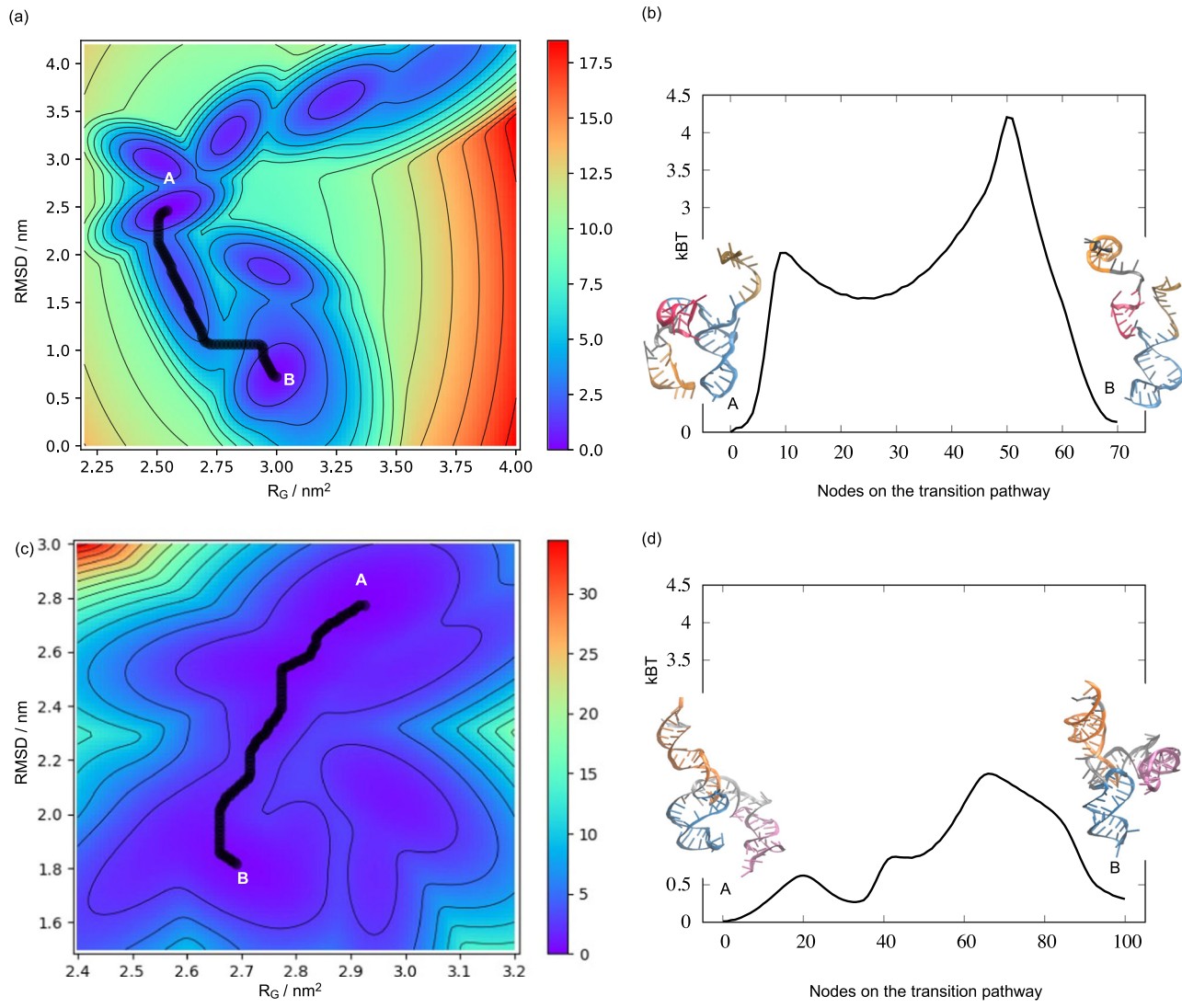

**Fig. 3 | Metadynamics simulations for refining the cryo-OrbiSIMS restrained RNA models. a, b** denote results for the sg RNA in the Cas9:sg complex and (**c, d**) for the 5s rRNA in the native ribosomal complex. **a, c** Free energy landscapes derived from the metadynamics simulations of the restrained 3D models (denoted as A) show that these models converge to a PDB-like structure (denoted as B) following a minimum energy transition pathway (black). **b, d** Energy trace of this minimum energy pathway shows that the transition barrier between the restrained

and the refined models is less than $4k_BT$ (where $k_B$ is Boltzmann constant and T is the temperature in Kelvin), and these states could thus represent the conformation dynamics of the RNA under physiological conditions. The restrained and refined structures are also denoted in cartoon representation and the corresponding helices, stems and loops in the two structures are coded using the same color scheme.

restraints had a pronounced impact on refining the modelling of the 3D structures[26–28] (Fig. 2b, d and Supplementary Table 3), as indicated by the improved TM-Score metric[29] of the restrained models when compared to the de novo prediction of the 3D structures. However, the 3D models still exhibited global RMSDs of more than 20 Å when compared to the respective PDB structures (Supplementary Table 3). On closer observation, we could attribute this discrepancy to an alternative orientation of a secondary structure element within the global conformation of the restrained 3D model. For instance, in the sg RNA, stem-loop 2 showed a 180° flip compared to the PDB structure (Fig. 2b, Supplementary Fig. 8a), and in 5s rRNA, Loop C has a similarly flipped orientation (Fig. 2d, Supplementary Fig. 9a).

To test if these orientations could be further refined to better represent their global conformations, we performed metadynamics simulations to investigate the likelihood of the cryo-OrbiSIMS restrained models converging to their corresponding PDB structures. For both the sg RNA and 5s rRNA, our results indicate that the initial cryo-OrbiSIMS restrained models converged to structures that

closely resemble the native PDB configuration (Fig. 3a, c and Supplementary Figs. 8b and 9b). Notably, the starting and the final conformations appear to represent local energy minima within the free energy landscape[30] of each RNA molecule, with a transition state barrier estimated to be less than $4k_BT$ (Fig. 3b, d). This suggests that the structural changes between these states are energetically feasible and could represent native-state dynamics of these RNAs. In contrast, the unrestrained model did not converge to the PDB-like structure (Supplementary Fig. 10). These results strengthen our confidence in the effectiveness of cryo-OrbiSIMS restraints for accurately modelling the 2D and 3D structures of RNA.

**Modelling structural plasticity of 7SK RNA using cryo-OrbiSIMS**
The 7SK RNA is a key component of the 7SK RNP (Supplementary Fig. 15), a master regulator of all gene transcription by RNA Polymerase II in the cell. It sequesters positive transcription elongation factor, PTEF-b, in an inactive form and several cellular and viral factors remodel the 7SK RNP structure to release PTEF-b and modulate gene

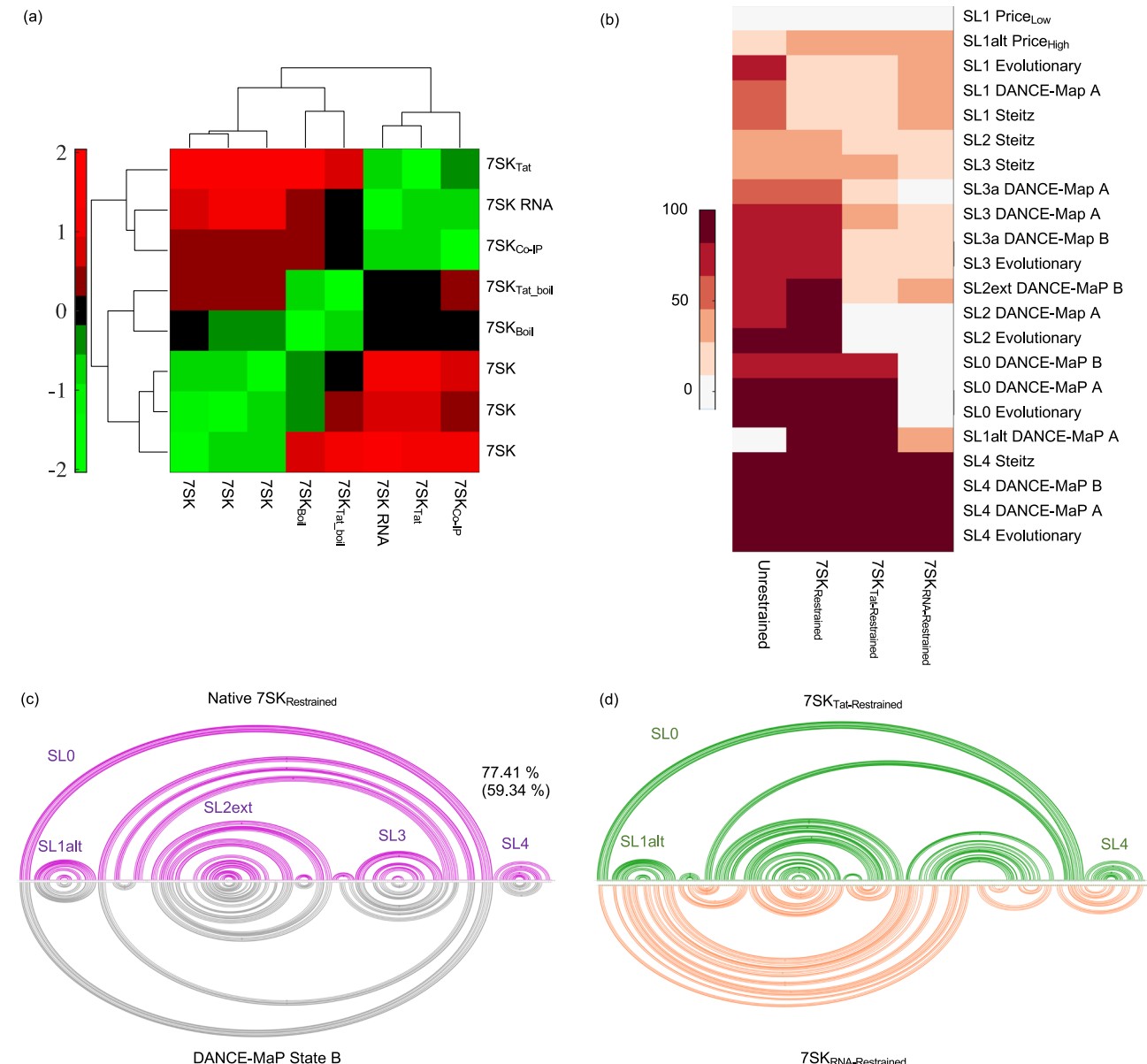

**Fig. 4 | Structural plasticity of the 2D folds of 7SK RNA validated using Cryo-OrbiSIMS. a** Hierarchical clustering of the negative polarity data for apo, native, denatured and Tat-remodelled 7SK denote that the 7SK RNA has distinct structural characteristics in these samples. **b** Percentage similarity of the secondary structure elements in the 2D fold of 7SK determined in this study with previously determined secondary structures of the RNA using DMS and SHAPE reactivity[31–34]. **c** The native restrained 7SK is found to resemble State B[31] conformation with 77.41% secondary structure similarity and 59.34% identity. **b**, **d** Similarly, apo restrained 7SK RNA structure, particularly in stem-loop 1, resembles the 7SK structure determined under low magnesium conditions previously[32]. Tat-remodelled 7SK seems to be a distinct conformation, consisting of SL0, SL1alt and SL4 features from State B, but other base-pairings in SL2 and SL3 distinct from any previous reports.

expression under various health and disease conditions. Despite rapid progress in our understanding of the molecular mechanism of 7SK RNP assembly and function, a high-resolution analysis of the structural plasticity of 7SK remains elusive, particularly due to challenges associated with high instability, conformational flexibility, and dynamic interactions of the 7SK components.

To bridge this gap, we employed the benchmarked cryo-OrbiSIMS pipeline to elucidate the 3D architecture of the 7SK RNA in three distinct states: its unbound apo form, its native configuration when bound to cellular proteins, and its altered conformation remodelled by HIV Tat protein. Analysing the negative polarity cryo-OrbiSIMS data for these samples through z-score clustering (Fig. 4a), we found that the apo 7SK RNA and the remodelled 7SK RNA cluster away from the native and denatured forms of 7SK, indicating significant structural differences of

the 7SK RNA within these forms. Furthermore, when the cryo-OrbiSIMS peak assignments were used as structural restraints to predict the 2D folds of the 7SK RNA (Fig. 4b–d and Supplementary Fig. 11), we observed that the structure of the native 7SK globally resembled the State B conformation recently elucidated by the DANCE-MaP algorithm[31] (Supplementary Table 2), and contained SL0, SL1alt, SL2ext, SL3 and SL4 domain characteristics with more than 50% base pairing identity with other previously reported structures[31–34]. Moreover, the restrained structure of the apo 7SK RNA, particularly in SL4 and SL1, also exhibited parallels with the architecture of the RNA documented in these reports. In contrast, the restrained structure of Tat-remodelled 7SK exhibited distinct features. While it comprised of the SL0, SL1alt, and SL4 stem-loops from State B, akin to a PTEF-b released form, it also exhibited several other base pairings within the SL2 and SL3 regions.

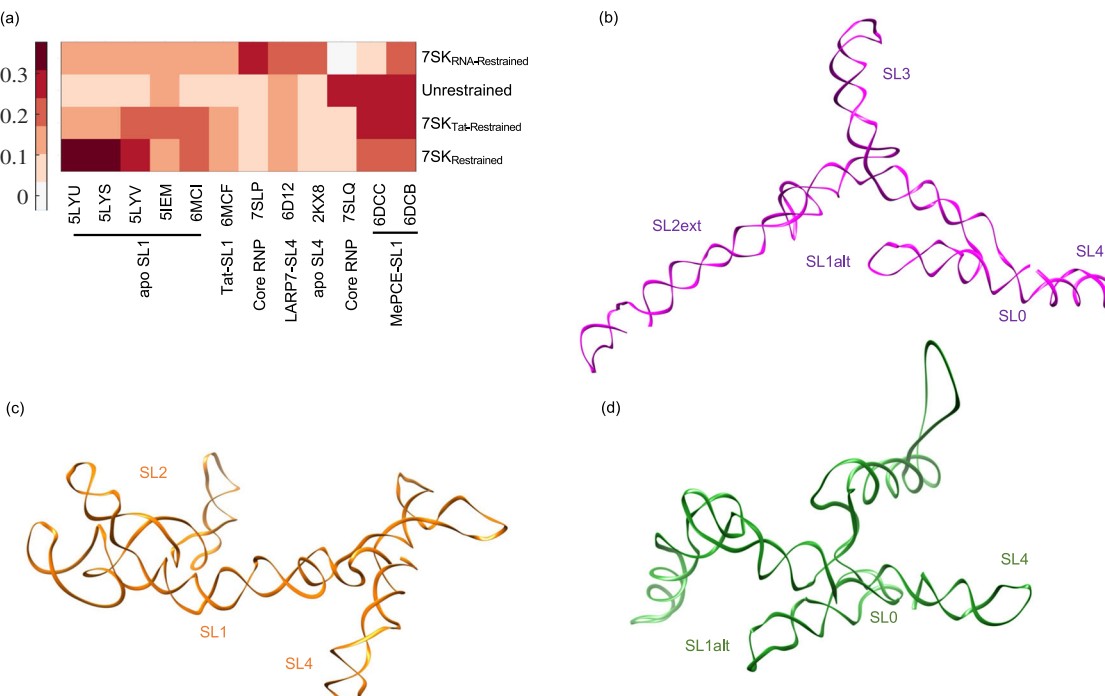

**Fig. 5 | Comparison of restrained 3D structures of the 7SK RNA in its apo, native and remodelled states. a** The modelled restrained structures share structural similarities with the PDB structures of SL4 and SL1 deposited in the PDB with low local RMSD values. **b–d** Global conformation of the restrained 7SK RNA in its (**b**) native state, **c** apo state and (**d**) Tat-remodelled state. The conserved stem-loop characteristics within these states are annotated on the 3D structure.

We then used the restrained 2D folds to model the 3D structure of the apo, native and remodelled 7SK RNA[25]. The resulting conformations for the 3′ stem-loop in our restrained native and remodelled 7SK demonstrated RMSD values less than 0.2 nm with the PDB entries for SL4 (Fig. 5a), and thus closely resembled experimentally determined SL4 structures in both its apo and protein-bound forms[35–37]. In contrast, the 5′ region of the restrained apo 7SK structure exhibited similarities to the PDB structures associated with SL1[38–40] of 7SK RNA. This consistency is supported by the observation that the restrained, protein-bound forms of 7SK in our study exhibited characteristics of SL1alt, while the restrained apo 7SK structure contained SL1, thus providing robust support for the reliability of our 3D structure predictions for the 7SK RNA.

## Discussions

The findings from this benchmarking study have evaluated the repeatability, sensitivity, and accuracy of cryo-OrbiSIMS for characterising RNA and RNA-protein complexes at high-resolution. The enhanced chemical coverage observed in cryo-OrbiSIMS data of native bacterial ribosomes aligns well with previous studies[10,12], which revealed that cryogenic conditions not only enhance ionisation yield but also reduce ion-beam-induced fragmentation of sensitive biomolecular samples[12]. Additionally, the frozen hydrated state maintained during cryo-OrbiSIMS data collection has been shown to enhance signal intensities for polar samples[10], such as RNA nucleotides, which has been particularly advantageous for our study. Furthermore, cryogenic conditions effectively preserve the native conformations of the investigated biomolecular complexes[41]. In contrast, preparing samples and collecting data at room temperature can lead to sample dehydration due to solvent vaporisation, potentially resulting in denaturation of biomolecules.

The exact mechanism by which the argon cluster ion beam induces ballistic fragmentation of RNA and its complexes in cryo-OrbiSIMS is unknown. The fragmentation in SIMS occurs through physical bombardment and transfer of kinetic energy from the argon

atoms to the analyte[42,43]. However, operating the OrbiSIMS in cryogenic conditions affects both sputtering[44] and ionisation of the molecules[10], and our results provide clear evidence of a complex relationship between the spectrum assignment frequency of an RNA residue and, by extension, its tendency for fragmentation, with its native contacts within the parent system. To arrive at a comprehensive interpretation of our cryo-OrbiSIMS data, we first ascertained that we were collecting data from the entirety of the sample (full-depth sputtering in Supplementary Fig. 16) and considered four possible scenarios (Supplementary Fig. 17) that could rationalise these experimental observations. Scenario A represents our initial hypothesis, where the exposed regions of the RNA are expected to be bombarded more and thus fragmented more, leading to a higher number of residue peak assignments from these regions. However, our peak assignment frequencies are not correlated to the residues' solvent accessible surface area (Supplementary Fig. 5), thus this scenario is unlikely. Alternatively (Scenario B), if the molecules are adsorbed on the gold-surface in a preferred orientation, the outer regions of the RNA systems will be fragmented more and would thus produce shorter fragments compared to the inner regions of the complex. However, we do not see any correlation between the average fragment length observed for an RNA residue and it's solvent exposed surface area (Supplementary Information Spreadsheet), suggesting that this scenario is also unlikely, and the RNA complexes might not have been preferentially oriented on the surface. In Scenario C, the full depth sputtering (Supplementary Fig. 16) means that all parts of the sample will be exposed to the beam at some point during the data collection process and should give rise to equal numbers of fragments from all regions (either exposed or buried) of the complexes. However, we clearly see an enrichment in spectrum from the RNA:RNA and RNA-protein interaction sites (Fig. 1e) and thus Scenario C is also unlikely. However, previous studies[45,46] have demonstrated that the secondary ion yields are higher from regions with higher cohesive energies, such as the soft interactions between the RNA-RNA and RNA-protein residues. Furthermore, such interactions would also stabilise the labile

phosphodiester backbone[47] of the RNA, likely producing specific fragments from these regions in the detectable mass range of the OrbiSIMS. Based on this understanding, we speculate that Scenario D is a likely case and propose it as our refined working hypothesis for the observed fragmentation of the RNA complexes in cryo-OrbiSIMS.

A number of experimental parameters can be modified to retain relatively large fragments from macromolecules and reveal additional information about the molecular structure using OrbiSIMS. These are respectively, changing the energy per nucleon by increasing the cluster size or reducing primary beam energy[48], and tuning of the target potential ($V_T$)[49]. Due to the multitude of factors affecting the results, tuning of these parameters and understanding their impact on molecular structure information obtained from the OrbiSIMS spectra is subject for future work.

The average mass of a ribonucleotide monophosphate is 339.5 Da, suggesting that the cryo-OrbiSIMS mass range of up to $m/z = 2250$ can effectively detect neutral losses corresponding to RNA fragments with a maximum length of six to seven nucleotides. To assess the applicability of this mass range in characterising biologically relevant RNAs, we derived seven-nucleotide fragments from the RNA complexes deposited in the Protein Databank and observed that these fragments could uniquely align to their own parent sequences, which could be up to 1870 nucleotides in length (Supplementary Fig. 12). Such sequences already account for more than 94% of the high-resolution (< 3 Å) RNAs and their protein complexes deposited in the PDB. Conversely, when we used a random seven-nucleotide sequence, it aligned to an average of 40 positions spanning 4.92 unique entries in the PDB. This signifies that a random sequence of seven nucleotides could be potentially matched to 8.13 peaks within the cryo-OrbiSIMS spectrum, highlighting the potential for redundancy in assignments. Such redundancy in assignments of the negative polarity data for RNAs is a result of the low chemical diversity in nucleic acid sequences, where only four types of nucleotides are present (compared to twenty different types of amino acids in proteins). Furthermore, nucleotide fragments of a given length $N$ can have $^4P_N$ possible sequences, all with the same mass and potentially aligning to the same peak. Consequently, the process of assigning peaks becomes more complex with higher $m/z$ and longer RNA systems studied, which may limit the practical application of cryo-OrbiSIMS to known RNA systems less than a couple of thousand nucleotides in length.

In our study, we observed a remarkably high correlation (R = 0.99) between the length of the RNA and the level of redundancy in peak assignments. However, the redundancy values for minimum RMSE assignments always positioned around the median value for the dataset (Supplementary Fig. 4). This suggests that the most accurate assignments, with the lowest RMSE, were not significantly affected by the degree of redundancy observed in the data. Yet, to potentially avoid complications from false positive assignments, we decided to exclude the 16s rRNA and 23s rRNA data (redundancy values of 23 and 36, respectively, and significantly higher than the PDB threshold of 8.13) from downstream structural analysis. By excluding such the systems with higher redundancy values, we aimed to enhance the accuracy of our data interpretation for this benchmarking study.

For 7SK, we observed that its unrestrained 2D fold, particularly through the presence of canonical stem loops 1 to 4, exhibited secondary structure elements that remarkably aligned with previous studies. This striking consistency is likely a result of the inclusion of 7SK as a training dataset in the development of the majority of RNA structure prediction algorithms. However, by integrating cryo-OrbiSIMS data as experimental restraints, we were able to deviate from this canonical 2D structure, and steer the two-dimensional folding of the 7SK RNA towards its native, predominant State B conformation[31].

The absence of high-resolution structures of full-length 7SK RNA poses a significant challenge in precisely determining the accuracy of the global conformations of the restrained 7SK in its unbound, native, and remodelled states. Despite this challenge, the tertiary structure prediction algorithm[25] was able to successfully preserve the local structural characteristics of SL1 and SL4, which have been well-studied in both their unbound and protein-bound forms. Furthermore, in alignment with our 2D predictions, the 3D model of the restrained native 7SK RNA exhibited all the distinctive structural features of State B[31]. Similarly, the 3D structures of the restrained apo 7SK maintained distinct SL1, branched SL2, SL3 and SL4 features; while the restrained Tat-remodelled 7SK exhibited SL0, SL1alt and SL4. Such consistency between the restrained 2D and 3D predictions highlight the fidelity of our approach in capturing the inherent structural plasticity of the 7SK RNA in its different forms. Moreover, these predicted models now could be reliably used in guiding the structure determination of the 7SK RNA with its various binding partners and under different biological contexts, thus providing a comprehensive understanding of its structural and functional heterogeneity.

Systems containing at least one or more RNA entities comprise less than 4% of the total number of structures deposited in the Protein Data Bank (PDB), with even fewer of these structures being resolved in their native forms. This low number can be attributed to challenges associated with obtaining RNA samples in yield, stability, and purity levels amenable for crystallisation or cryo-EM investigations. Our results emphasise that the integration of cryo-OrbiSIMS with computational modelling tools has the potential to close this gap and enable a more accurate representation of native RNA conformations to understand their structure-function relationships. Such insights could help life sciences researchers to understand more broadly how RNA regulates gene expression, how it is involved in diseases, and how we can design new drugs that target RNA.

## Methods
### RNA purification and sample preparation
Ribosomes were purified from *E. coli* by adapting a method previously described by Blaha and co-workers[50]. The sample purity (Supplementary Fig. 13) was analysed via negative stain Transmission Electron Microscopy. The purified ribosomes were concentrated to 600 OD, flash frozen in 5 μL aliquots in liquid nitrogen and stored at −80 °C until further use. For the OrbiSIMS experiments, the samples were analysed at a final concentration of approximately 7.5 nM, which was obtained by diluting the purified ribosomes in Buffer A (20 mM Tris, pH 7, 100 mM ammonium chloride, 10.5 mM magnesium acetate, 0.5 mM EDTA and 5 mM β-mercaptoethanol). For denaturation, the native ribosome samples were boiled at 95 °C for 5 mins.

Recombinant Cas9 protein was purchased as a lyophilised powder from Sigma-Aldrich (Catalogue No. CAS9PROT-250UG) and reconstituted in 25 μL of 50% glycerol in water according to the manufacturer's instructions. The 85 nucleotides long single guide (sg) RNA sequence (Supplementary Fig. 14) was obtained from Protein Databank Entry ID 4ZT9, which describes the structural investigation of Cas9:sgRNA interactions[17]. This canonical sequence was mutated (1) from $A_{46}A_{47}A_{48}A_{49}$ to $C_{46}C_{47}C_{48}C_{49}$ to disrupt the crRNA:tracrRNA interaction and (2) from $A_{51}A_{52}G_{53}G_{54}$ to $C_{51}G_{52}U_{53}U_{54}$ to destabilise stem loop 1 in the tracr RNA (Supplementary Fig. 14). The structures of the canonical and mutant sg RNAs were predicted using RNAComposer[24] to ascertain the extent of structural rearrangement of the canonical sg RNA upon introduction of mutants. Cas9:sg complex was prepared by in vitro reconstitution where the Cas9 protein solution and the sg (wild type or mutant) RNA solution was mixed at a 1:1.6 ratio and final concentrations of 10 nM and 16 nM respectively, and incubated at 4 °C for 30 mins.

Native 7SK RNP was purified from HeLa cell extract using co-immunoprecipitation with anti-HEXIM antibody (Proteintech, 15676-1-AP) and biotinylated DNA primers (5′-[Btn]ACCTTGAGAGCTTGTTTGGA GG)[51]. Co-immunoprecipitation with anti-HSP90 (Proteintech, 13171-1-AP)

was used as negative control. Briefly, 100 µl of the cell extract at 2 mg/ml total protein concentration was incubated with the primer (10 µM final concentration) or the antibody (at supplier recommended concentration) overnight at 4 °C and for a further 1 hour at room temperature and the 7SK complex was pulled down using streptavidin or Protein A beads respectively. The bound complex was eluted using a reverse complement 7SK primer (5′-CCTCCAAACAAGCTCTCAAGGT) at a final concentration of 50 µM or boiling the beads in binding buffer. Unlike the co-IP elution, biotinylated primer pull-down ensured that 7SK RNP could be eluted from the streptavidin beads under physiological conditions and could be maintained in its native conformation. The sample composition was verified by Western Blotting analysis, Mass Spectrometry, and reverse transcription PCR (Supplementary Fig. 15). Western blots were performed using primary antibodies against CDK9 (11705-1-AP) and Hexim1 (15676-1-AP) from Proteintech. Secondary antibody was goat anti rabbit conjugated to HRP (ab205718, Abcam) and blots were visualised using ECL solution and a Bio-Rad Chemidoc MP imager (Bio-Rad Laboratories, Watford, UK). HIV-1 Tat protein was recombinantly expressed and purified from *E. coli* BL21 cells as previously described[52]. For remodelling the 7SK RNP structure, purified Tat (10 µM final concentration) was added to the incubation of cell extract with biotinylated primers. For generating negative control samples, *E. coli* cell lysate instead of HeLa cell extract was used in the 7SK purification procedure described above.

7SK RNA was obtained by in vitro transcription from the canonical 7SK gene sequence cloned into pUC19 plasmid vector (GenScript Biotech Corp, NJ, USA) using HiScribe® T7 Quick High Yield RNA Synthesis Kit (E205S, New England Bio-labs, MA, USA) and T7 Ribo-MAX™ Express Large Scale RNA Production System (P1320, Promega Corp., WI, USA) according to manufacturer's instructions. The 7SK RNA transcripts were cleaned with Monarch® RNA Clean-up Kit (50 µg) (NEB, MA, USA), quantified by Qubit 4 Fluorometer RNA BR Assay (Thermo Fisher Scientific, MA, USA) and band quality was analysed by 4200 TapeStation System (Agilent Technologies Inc, CA, USA) at the DeepSeq facility (University of Nottingham, Nottingham, UK).

## OrbiSIMS sample preparation

The samples were absorbed on a gold substrate by spotting 3 uL sample drop on a gold-plated glass slide. The spot was incubated for 45 s before excess sample was blotted off using a piece of Whatman paper. The sample spot was then washed 3 times with phosphate-buffered saline (PBS), blotting away any excess liquid. For room temperature data collection, the substrate was allowed to air dry before OrbiSIMS measurement. For cryogenic conditions, samples absorbed and washed on the substrate were immediately flash frozen by plunge freezing the substrate in liquid nitrogen and maintained under liquid nitrogen temperatures during storage and data collection.

## OrbiSIMS data collection

All OrbiSIMS data was collected under cryogenic conditions at −150 °C, except for *E. coli* ribosomes, where sample preparation and data collection were additionally performed at room temperature. For the cryogenic measurements, the VCM/VCT cryotransfer system and the OrbiSIMS temperature control system took about an hour to reach the required temperature (< 150 °C). For each sample, data was collected in three replicates, each in the negative and positive polarity. Two replicates (negative and positive polarity each) were also collected for the gold substrate and were used as reference spectra for data processing. Due to the adsorption of the samples as a thin layer, each measurement took a couple of minutes to complete before the whole sample got sputtered away by the incident argon beam (Supplementary Fig. 16). Mass calibration of the Q Exactive instrument was performed once a day using silver cluster ions. Electron flood gun and argon gas flooding were used for charge compensation and the target potential was set to −57 V in negative polarity and +57 V in positive

polarity. Single beam 20 keV $Ar_{3000}^+$ with Orbitrap Q Exactive analyser were used for all measurements and the current of the primary beam was 230 pA, total ion dose was $3.65 \times 10^{13}$ ion/cm² and probed surface area was 200 × 200 µm. For all Orbitrap data, mass spectral information was collected in a range $m/z$ = 150 − 2250. The mass resolving power of the Orbitrap analyser was set to 240,000 at $m/z$ = 200.

## Data processing

IonToF SurfaceLab 7 was utilised to extract peak lists from the raw spectra. Subsequent analysis of the peak lists was carried out using MATLAB 2021a[53] and BASH shell scripting. For the gold substrate data, the common peaks between the two replicates for each polarity were retained as a reference spectrum. For the RNA samples, the replicate peak lists were processed in pairs to eliminate (1) non-overlapping peaks between the replicate data and (2) overlapping peaks with the reference spectrum of the corresponding polarity. During the peak comparison, a tolerance of 1 ppm was applied to account for any $m/z$ value variations between the replicate data. When two replicates were available, a single final peak list was generated for subsequent analysis. However, if three replicates were measured, three processed peak lists were obtained by comparing replicas 1 versus 2, replicas 2 versus 3, and replicas 1 versus 3. The processed peak lists were used for Wasserstein Distance calculation and for the Kruskal-Wallis test performed using MATLAB 2021a.

## Peak assignments

Peak assignments were performed by comparing the experimental $m/z$ values to a database of theoretical predictions for all the possible fragments from the given RNA sequence. A 2 ppm error and an additional difference of phosphate, hydroxyl or proton masses were also considered during the comparisons. The theoretical predictions were performed on RNA sequence fragments generated using sliding window sizes of 4 nucleotides and longer. This compensated for the computational cost of creating the theoretical database over the full-length sequence, which also exponentially increased with the length of the RNA studied. For each window fragment, the $m/z$ values were calculated for each type of RNA ions, "a" to "d" and "w" to "z" ions (Supplementary Fig. 18). For each window, the cumulative error for each assigned peak was summed up to derive the root mean squared error (RMSE) for each window and was then averaged over the total number of windows assigned in the full-length RNA. To narrow down the final assignments, the assignment data with the lowest RMSE was utilised, as it most closely matched the experimental spectrum. During the peak assignment process, the redundancy of assignments was also quantified by calculating the average number of times a specific cryo-OrbiSIMS peak was assigned to multiple, distinct ions fragmented from the corresponding parent RNA.

To quantify the acceptance threshold for the redundancy, we simulated peak assignments for the complete database of RNA sequences derived from high-resolution (<3 Å) Protein Data Bank (PDB) structures. This database contained a total of 3717 sequences with a multimodal distribution of lengths ranging from 4 to 5070 nucleotides (Supplementary Fig. 12). Each entry in the database was fragmented into child sequences ranging from 1 to 50 nucleotides in length and each child fragment was aligned to its corresponding parent sequence to identify potential redundancy in the alignments. As anticipated, a positive correlation was observed between the length of a child fragment and its ability to uniquely align with its parent RNA (Pearson Correlation Coefficient, R = 0.73). Next 50 random, seven-nucleotide long fragments were also generated, and compared against all entries in the RNA dataset, which seemed to align with an average of 40 positions across 4.92 unique entries (i.e., 8.13 positions in each entry) in the dataset. Thus, the acceptance level for the redundancy was set to 8.13 for assessing quality of the peak assignment algorithm.

## Principal component analysis

Principal component analysis was performed on the sg RNA and 5s rRNA systems to understand the relationship between the following four characteristics of their RNA residues: (1) frequency of assignments in the cryo-OrbiSIMS data, (2) native contacts, (3) base-pairing and (4) solvent accessible surface area (SASA). The number of RNA-RNA or RNA-Protein native contacts and SASA for the RNA residues in the corresponding RNP complex (PDB IDs 4ZT9 and 6GWT) were calculated using GROMACS 2021.5[54] functions gmx select and gmx sasa respectively. Base pairing of the RNA residues was indicated in binary notation (0 = unpaired or 1 = paired) and determined from the corresponding PDB structures (IDs 4ZT9 and 6GWT). PCA was computed on this data using MATLAB 2021a.

## Structure modelling and simulations

RNA secondary structure (2D fold) prediction was carried out using the RNA structure Fold[15,16] algorithm. This algorithm predicts the lowest free energy conformation for a given RNA sequence and can utilise experimental data as restraints to guide the folding process. The frequency of peak assignment for each RNA residue in the cryo-OrbiSIMS spectrum was employed as base-pairing restraints to bias the folding process towards conformations that were more likely to be consistent with the cryo-OrbiSIMS data, resulting in a refined and more accurate prediction of the RNA secondary structure. The tertiary structure (3D conformation) of the RNA was predicted using the RNAComposer[24,25] webserver. This takes as input the RNA sequence and the 2D fold predicted by RNAstructure. Secondary structures were compared using the Beagle (BEar Alignment Global and Local) algorithm[55], which quantified the percent similarity and identity between secondary structures provided. For comparing tertiary structures, US-align[29] was used to compute the RMSD and TM-Score metric for global RNA structure comparison. Chimera Matchmaker[27,28] was used to visualise the 3D structures.

For 3D structure refinement, metadynamics simulations were performed as previously described[56]. All simulations were performed in GROMACS 2021.5[54] using the AMBER99 OL3 force field[57] with DE Shaw parameterisation[58]. The restrained structures were used as starting conformations and placed in a dodecahedron box with a distance of 12 Å between the sides of the box and the molecule. TIP4EW water model was used to solvate the system, and the charge was neutralised by adding 42 $Mg^{2+}$ ions to the sg RNA simulation box and 60 $Mg^{2+}$ ions to the 5s rRNA simulation box. The simulation protocol consisted of a steepest descent energy minimisation followed by a low-memory Broyden–Fletcher–Goldfarb–Shannon quasi-Newtonian minimiser. Subsequently, a 50 ps simulation at 200 K was performed without pressure coupling, with position restraints applied to the RNA. The position restraints were then removed, and the temperature of the system was raised to 298.15 K for an additional 100 ps simulation under NVT conditions. Afterwards, pressure coupling was activated, and the system was equilibrated under NPT conditions.

Replica exchange metadynamics simulations were performed on the equilibrated system. Metadynamics enhances sampling of rare or transition states in molecular simulations. It achieves this by applying a history-dependent bias potential on specific degrees of freedom of the system, known as collective variables. These bias potentials help the system overcome energy barriers, enabling exploration of various conformations and transition pathways. For the sg RNA, the instantaneous RMSD of the simulated structures from the 4ZT9 PDB structure[17] was used as a single collective variable. For the 5s rRNA, the relative angles between the helices in the structure were used as the collective variables. During the simulations, the metadynamics trajectories were allowed to exchange with a neutral replica, which served as a reference for measuring the convergence of the simulations to the canonical distribution associated with an unbiased potential of the system. This exchange with the neutral replica ensured the reliability and accuracy of the simulations in exploring the system's energy landscape.

## Reporting summary

Further information on research design is available in the Nature Portfolio Reporting Summary linked to this article.

## Data availability

The raw cryo-OrbiSIMS data generated in this study is deposited in The University of Nottingham Collection of Public Research Data[59] [https://doi.org/10.17639/nott.7354] and can also be made available upon request from the corresponding author. All processed data supporting the findings of this study is provided within the main text and supplementary information. The following Protein Data Bank entries were used to benchmark the cryo-OrbiSIMS modelled 2D and 3D structures: (1) PDB ID 4ZT9 for Cas9:sg complex, (2) PDB ID 6GWT for bacterial ribosome complex, and (3) PDB IDs 5IEM, 5LYS, 5LYU, 5LYV, 6MCI, 6MCF, 6DCB, 6DCC, 2KX8, 6D12, 7SLP and 7SLQ for the 7SK RNP complex. Additionally, DMS probing dataset [https://doi.org/10.7554/eLife.22037] and IM7 probing dataset from RMDB 5SRRNA_IM7_0006 [https://rmdb.stanford.edu/detail/5SRRNA_1M7_0006] were used for performance comparison of cryo-OrbiSIMS with these techniques.

## Code availability

Custom scripts written for MATLAB 2021a were used to perform all the data analysis. These script files along with a comprehensive code documentation file that enlists the system requirements, installation instructions, step-by-step instructions for running the scripts and explanations of file formats and functionalities, and a sample dataset for users to test and understand the code's functionality are available on GitHub repository https://github.com/BorkarLab/OrbiSIMS_RNA_analysis/tree/RNA_published[60] with https://doi.org/10.5281/zenodo.10960751.

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

## Acknowledgements

The authors would like to thank UKRI Medical Research Council IMPACT Doctoral Training Programme funding to SW (MR/N013913/1), The Alan Webster Endowment funding for CW, the School of Veterinary Medicine and Science, the University of Nottingham funding to AC, Anne McLaren Fellowship granted by the University of Nottingham to ANB, BBSRC Discovery Fellowship (BB/S011102/1) to RB, and the Engineering and Physical Sciences Research Council (EPSRC) grant code EP/P029868/1, which funded the OrbiSIMS instrument at University of Nottingham. ANB would like to thank Dr Matthieu Gagnon, University of Texas Medical Branch, USA, for providing the *E. coli* ribosome samples used in this study. The authors thank Dr David Scurr, University of Nottingham and Dr Jeetender Chugh, Indian Institute of Scientific Education and Research Pune, India, for helpful discussions.

## Author contributions

Conceptualisation: A.N.B., A.M.K., Methodology: A.N.B., A.M.K., J.A.W., Investigation: A.N.B., C.S., S.W., C.W., A.C., R.B., Visualisation: A.N.B., A.C., R.B., Funding acquisition: A.N.B., A.M.K., Project administration: A.N.B., Supervision: A.N.B., Writing – original draft: A.N.B., R.B., Writing – review & editing: A.N.B., R.B., A.M.K., J.A.W. and S.W.

## Competing interests

The authors declare no competing interests.
