## [Peer Review File · Nature Communications]

Integrating Cryo-OrbiSIMS with Computational Modelling and Metadynamics Simulations Enhances RNA Structure Prediction at Atomic ResolutionREVIEWER COMMENTS

Reviewer #1 (Remarks to the Author):

The authors present an interesting study using mass spectra from cryo-OrbiSIMS as restraints in metadynamics simulations to determine the 3D structure of native structures of RNA. Comparisons with the PDB structures are provided. Their method is applied to 7SK RNP in the apo, native-protein bound, and modified states. This is of significant interest to the community. The study has been conducted carefully with good controls for validation. The experimental method is described well with enough detail that would allow replication in another laboratory.

1. Cryo-OrbiSIMS is quite a time-consuming experiment (it would be useful to note in the methods how long it took to acquire the data) to what extent can similar data be achieved using more conventional methods such as LC-MS? Can some guidance be provided when Cryo-EM or Cryo-OrbiSIMS would be most appropriate?
2. It has been found that there is structure (doi.org/10.1002/sia.7058) in the signal intensity vs the sample target potential, VT. In this study, standard operating values of +/- 57 volts have been used. Whilst the reviewer appreciates that this is a communication rather than an expansive study, it would be very interesting to include VT scans for the RNA ions. Perhaps this could extend coverage and or sensitivity.
3. In this study, 20 keV Ar3000+ is used giving ~ 6.7 eV per atom, which is quite high. Since no imaging is needed and a good focus is not necessary then a lower beam energy could be used, for example 5 keV Ar2500+. According to ref (12) this is likely to reduce fragmentation further and may increase the length of sequence that can be detected. It would be helpful to include some supplementary data on this.
4. Line 222 – “Closely resembled the PDB structures” is rather subjective. Can a more quantitative metric be used for closeness?
5. Figure 1 – Please explain how the data has been scaled prior to PCA analysis. The first principal component seems to describe almost all the variation with little information in the second and third. Please provide some discussion on what this means.
6. Figure 3 - Please check it looks like Fig 3 (b) and Fig 3 (d) have got switched as they do not match the topology in Fig 3 (a) and Fig 3 (c) with respect to the transition paths

Minor comments:

Line 120 – The arguments here are similar to those developed by David Castner to probe the orientation of surface bound proteins using ToF-SIMS (e.g. PMID: 15801458). It would be appropriate to add a reference.

Line 230 – what is meant by reproducibility here? Does it mean that the presented method reproduces structure measurements obtained using other methods or is it to do with the repeatability of the measurements in this study?

Line 223 – Note that the cluster only carries 1 charge and so it should be argon atoms not ions. Also, note that “argon” should be lowercase throughout.

Line 447 = note the sample target potential will be + 57 V for positive ions and – 57 V for negative ions.

Reviewer #2 (Remarks to the Author):

The article of Borkar et al. is groundbreaking because, by combining methods of structural biology and solid-state mass spectrometry, it manages to propose reasonable 3D structures for RNA and their complexes in the native environment. To my knowledge, it is the first time that SIMS data are directly used to establish the full 3D structure of complex biomolecules. Very cleverly, the authors use the frequency of peak assignment of the RNA residues in the cryo-orbi-SIMS spectra as base pairing

restraints to obtain a better prediction of the secondary and hence the tertiary structure. Finally, they convincingly calculate the structure of 7SK RNA in its native state, apo state and remodeled by HIV Tat protein. The research is ambitious and the results introduce a step change for biological SIMS but also a very interesting evolution for structural biology. The amount of work is very impressive and the authors selected only the most salient results to show in the article, the rest being provided in SI (including a full list of identified ions, useful for researchers in the field). For these reasons, the paper deserves to be published in Nature Comm after revision.

I have some comments, questions and requests for clarifications:

121 "Thus, flexible, unbound, or exposed regions are expected to experience greater fragmentation"
124 "we proposed a hypothesis that the number of times a particular residue is assigned within different ionised fragments of the sample, and subsequently its association with multiple peaks in the cryo-OrbiSIMS spectrum, directly corresponds to its exposure or burial within the studied RNA system."

These sentences explain the working hypotheses of the authors, at the core of this study, and those might need to be somewhat refined.

First, it is known that 20 keV Ar3000+ clusters will create in organic and molecular solids craters that are of the order of 10nm wide and 5nm deep. Therefore, they might be able to probe the molecules/complexes deeper than what the authors envision. It's been recently shown that clusters with those energies can desorb entire proteins and clusters of proteins with masses >50 kDa (intact or fragmented).

Second, the "depth" probed by the beam will depend on the cluster ion dose. In Orbi-SIMS, the dose used to obtain a sufficiently intense spectrum can be large, meaning that conditions are not static and notable erosion of the surface occurs. In that case one could expect that the entire biomolecule/complex is probed, and then the inside should show as much as the outside.

Unfortunately, I could not find in the method section the ion dose used for data acquisition, which would give an indication of the sputtered depth. The authors should give the ion doses in ions/cm² to help clarify that point and discuss it further if needed.

Third, the orientations of the molecules in the solid ice matrix are probably random with respect to the incoming beam direction, meaning that a mass spectrum acquired on a A x B μm² area (the probed surface area should be mentioned in the experimental section) might represent all the possible orientations. However, maybe it does not, because molecules will orient preferentially at the air water interface when the sample is created. If that's the case, orientation could affect the detected signals, over or under represent some parts of the molecules.

Fourth, if it is clear to me that higher sample cohesive energies (larger binding forces) lead to less sputtering (in general), I'm not at ease with discussing effects of "flexibility" etc. Intuitively I would think that "more condensed" parts (more native contacts, crystallinity, etc.) of the molecules would produce smaller fragments while longer and less bound chains would result in larger fragments, as seen with more "model" samples, which is a slightly different situation.

The fact that the authors find that their results contradict their hypothesis (negative correlation with SASA and positive correlation with number of native contacts) seems to indicate that the working hypothesis should be partly reformulated or at least refined. Perhaps the assignment frequency is not the right metric to correlate to the chosen parameters or the interpretation of this correlation is more complex.

Page 7. I don't understand the PCA analysis results. PCA on SIMS spectra should give scores and loadings corresponding to the different PCs, which can then be interpreted (or not) in terms of specific properties/features of the samples. In Figure 1, the authors directly name the axes of the principal components in the score plots as a representation of native contacts, assignment frequency, base pairing or SASA. What does it mean? In addition, the 3D plots of Figure 1 do not give any information because of the scale and the grouping of the points too close to one of the axes. The authors should show examples of the scores and loadings plots and explain how they reach their conclusions. They must explain the detailed procedure in the methods section. In general, the methods section lacks

important details, both for the experiments and the data processing.

Page 8. Generally, the method to retrieve the 3D structure is original and quite convincing. However, since it is a new method development, it seems particularly important to verify the results of the protocol (at least in some cases) with an independent method. I'm afraid it is somewhat lacking here but I agree that it might be a challenging task.

242 "The exact mechanism by which the Argon cluster ion beam induces ballistic fragmentation in cryo-OrbiSIMS is unknown. It has been recently reported that the fragmentation occurs through physical bombardment and transfer of kinetic energy from the Argon ions to the analyte³⁶." In fact, the mechanism of fragmentation induced by large Ar clusters, and its strong dependence on the energy per atom is quite well known now, because of the many experimental and computer simulation studies that have been published over the last 15 years, including the very recent study on proteins mentioned by the authors. The authors should reformulate the sentence and actually the whole paragraph. Beyond fragmentation, quite well studied, ionization remains a big question mark and the prevalent role of water, indicated by the difference between CRYO and RT SIMS, poses questions. The fact that SIMS may remain blind to certain species because they don't ionize well should not be overlooked.

Minor points:

81 while retaining the mass accuracy...

86 "These results are consistent with better ionisation yields¹² in cryo-OrbiSIMS compared to room temperature conditions due to hydrogen donation from H₃O⁺ and due to molecules retaining their native state¹³"

Their could also be a direct effect of the higher internal energy of the molecules at RT which adds to the energy brought by the impinging cluster.

Yet the percentages of common peaks between cryo and RT data is extremely low, which certainly points to a major role of the water matrix in the desorption/ionization process.

96 "These findings strongly indicate that cryo-OrbiSIMS measurements are reproducible"
Can the authors comment also on the reproducibility of the RT measurements?

109 "However, in the negative polarity data (Fig 1b and Supplementary Figure 2), the denatured ribosome sample clustered alongside the apo sg RNAs, separate from the native ribosome sample." Here, one could question whether the different results are due only to the denaturation or because of other differences in the initial sample. For example, are the quantities of frozen water between the denatured and native ribosome the same? Could there be an effect of the substrate (what is the sample thickness, I assume much larger than the cluster interaction volume)? Another fundamental question relates to the exact mechanism and timing of the interaction, which depends on the size and energy of the impinging cluster. For instance, do the molecules denature before breaking (unimolecular dissociation) or break immediately before desorbing ("ballistic" effect). The interpretations of the authors point to the second case, since denatured and native ribosome clearly separate. It is reasonable given that ~6 eV / Ar atom is already very energetic. The situation might be different with 1 eV/atom of energy.

187 "This suggests that the structural changes between these states are energetically feasible and could represent native state dynamics of these RNAs."

I would take a closer look into the number of hydrogen bridges. Have pair distribution functions of the elements.

247 "Thus, there may be physico-chemical processes influenced by local chemical environment of the analyte that may contribute to the observed fragmentation patterns,"

Very vague sentence. Maybe not necessary or should be rephrased.

Figure 3. Please show the minimum energy transition pathway in (a) the same way it is done in (c).

518 “the charge was neutralized by adding the appropriate number of Mg²⁺ and Cl⁻ ions.”
The explicit amount should be mentioned in the methods section.

Reviewer #3 (Remarks to the Author):

This work proposes Cryo-OrbiSIMS, a new mass-spec based approach to infer the number of contacts and flexibilities of each nucleotide residue in the RNA. While this information cannot be directly used for 3D structure modeling, it can be used improve RNA secondary structure prediction, which in turn improves RNA tertiary structure prediction.

My major concern about this work is its utility given the wide adoption of SHAPE-MaP, which, by all means, is the most commonly used experimental method to assist secondary structure determination. Both SHAPE-MaP reactivity and OrbiSIMS assignment frequencies measure the rigidity of the backbone, and they are both used (and only used) in RNA secondary structure determination. Given this fact, this work should explain whether OrbiSIMS is advantageous to SHAPE-MaP in terms of labor time and monetary cost. If not, is OrbiSIMS at least complementary to SHAPE-MaP for the task of secondary structure determination? For this purpose, the following should be implemented:

1. In the introduction, include SHAPE-MaP compare it to OrbiSIMS.
2. For the case study on Cas9:sg RNP and 5s rRNA, present the secondary structure calculated using SHAPE-MaP, using OrbiSIMS, and (optionally) using a combination of SHAPE-MaP and OrbiSIMS. Calculate the similarities between secondary structure derived from SHAPE-MaP/OrbiSIMS and that assigned based on experimental 3D structure by DSSR and/or RNAViewer. This benchmark is critical to evaluate whether OrbiSIMS is more suitable for the specific task of secondary structure determination.

There are also several minor technical issues that are worthy of additional revisions.

3. The title of this paper starts with Cryo-OrbiSIMS but the abstract only introduces 3D-OrbiSIMS. Please explain what the difference and relation are between the two in the abstract.
4. For Fig S5, in addition to current panels, it would also be helpful to add corresponding scatter plots, where each point is one residue, the x-axis is the structural feature (SASA or native contacts), and the y-axis is assignment frequency. Such a scatter plot will make the correlation (or lack hereof) more evident.
5. Chimera Matchmaker is probably not the best tool to evaluate RMSD between predicted 3D structure model and native (i.e., experimental) 3D structure. This is because Matchmaker does not guarantee the correct residue level correspondence between the model and native, and residue 1 in the model can be aligned to residue 10 in the native. To address this issue, the author can use the RNA_assessment tool provided by the RNA-puzzle assessors https://github.com/RNA-Puzzles/RNA_assessment. Alternatively, they can use US-align, which has an option called “-seq” that ensures pairwise sequence correspondence.
6. The title of this paper “Cryo-OrbiSIMS Enables Integrative Modelling of Native Structures of RNAs at Atomic Resolution” is highly exaggerated. Cryo-OrbiSIMS cannot be used for 3D structure modeling. It can only be used for secondary structure modeling. Even if a better secondary structure can indeed improve 3D structure modeling, the authors should not claim that Cryo-OrbiSIMS “enables” modeling of atomic structure.

Response to Reviewer Comments

We thank all three reviewers for providing positive feedback on our manuscript and for their valuable inputs to further improve it. We have now addressed all of their comments in full, please see the point-by-point response below.

Reviewer #1

Overall feedback: The authors present an interesting study using mass spectra from cryo-OrbiSIMS as restraints in metadynamics simulations to determine the 3D structure of native structures of RNA. Comparisons with the PDB structures are provided. Their method is applied to 7SK RNP in the apo, native-protein bound, and modified states. This is of significant interest to the community. The study has been conducted carefully with good controls for validation. The experimental method is described well with enough detail that would allow replication in another laboratory.

We highly appreciate the reviewer's encouraging feedback and thank them for recognising the significance of this work for the community.

1. Cryo-OrbiSIMS is quite a time-consuming experiment (it would be useful to note in the methods how long it took to acquire the data) to what extent can similar data be achieved using more conventional methods such as LC-MS? Can some guidance be provided when Cryo-EM or Cryo-OrbiSIMS would be most appropriate?

We agree with the reviewer that cryo-OrbiSIMS experiments can be time-consuming compared to the measurements at ambient temperatures. The most time-consuming process for our cryo-OrbiSIMS experiments was setting up the instruments in cryogenic conditions. For the cryogenic measurements, the VCM/VCT cryotransfer system and the OrbiSIMS temperature control system took about an hour to reach the required temperature (<150°C). However, once the systems were ready, each measurement took only a couple of minutes. This information has now been added to Methods subsection "**OrbiSIMS data collection**". e

The cryo-OrbiSIMS data measurement is relatively quick because the samples are deposited as a thin layer on the surface and the whole sample gets sputtered away within a handful of scans. To demonstrate this, the depth profile of representative 7SK sample and example gold ions as a function of primary ion dose has now been added to **Supplementary Figure 16**.

We appreciate the reviewer's query on the complementarity between LC-MS / Cryo-EM and Cryo-OrbiSIMS. A brief description has now been included in Supplementary Notes.

2. It has been found that there is structure (doi.org/10.1002/sia.7058) in the signal intensity vs the sample target potential, VT. In this study, standard operating values of +/- 57 volts have been used. Whilst the reviewer appreciates that this is a communication rather than an expansive study, it would be very interesting to include VT scans for the RNA ions. Perhaps this could extend coverage and or sensitivity.

We appreciate the reviewer's comment and agree that altering the VT can uncover additional information about the analyte as described by Matjacic et. al. However, VT scans require a thick, homogenous sample. In our study, the samples are adsorbed on the gold surface as a thin layer, which gets sputtered away completely within a handful of scans (**Supplementary Figure 16**) and thus, would be insufficient for a VT scan.

3. In this study, 20 keV Ar3000+ is used giving ~ 6.7 eV per atom, which is quite high. Since no imaging is needed and a good focus is not necessary then a lower beam energy could be used, for example 5 keV Ar2500+. According to ref (12) this is likely to reduce fragmentation further and may increase the length of sequence that can be detected. It would be helpful to include some supplementary data on this.

We acknowledge the reviewer's input regarding the effect of eV per nucleon on sample fragmentation. In our previous work with protein complexes larger than 50kDa in size (unpublished), we have observed that using larger argon cluster sizes (e.g., Ar5000+) or reducing the beam energy may diminish the signal obtained from such large macromolecules. Thus, due to the large size range of the RNA systems investigated in the current study, we have decided to use the parameters previously optimised for bio-macromolecular analysis by Scurr and co-workers (Kotowska, 2020).

1. Kotowska, A. M. et al. Protein identification by 3D OrbiSIMS to facilitate in situ imaging and depth profiling. Nat Commun 11, 5832 (2020).

4. Line 222 – “Closely resembled the PDB structures” is rather subjective. Can a more quantitative metric be used for closeness?

We thank the reviewer for this suggestion. The RMSD value for these structure comparisons has now been added to the main text at Line 232.

5. Figure 1 – Please explain how the data has been scaled prior to PCA analysis. The first principal component seems to describe almost all the variation with little information in the second and third. Please provide some discussion on what this means.

The PCA analysis in Figure 1c and 1d are not the loadings of the m/z values on the principal components as typically presented in SIMS data analysis. The PCA analysis instead is used to understand the relationship between the following four characteristics of each residue in the studied RNA: frequency of assignments, native contacts, base-pairing and solvent accessible surface area (SASA). The data wasn't scaled prior to PCA and as the reviewer has correctly pointed out, this analysis shows that the first principal component, corresponding to the native contacts of the residues, accounts for almost all the variation in the data. To make it clear, a discussion of what this means for the cryo-OrbiSIMS peak assignment analysis has been included between Lines 148 – 166 in the main text and a description for performing the PCA has been added to Methods subsection “*Principal Component Analysis*”.

6. Figure 3 - Please check it looks like Fig 3 (b) and Fig 3 (d) have got switched as they do not match the topology in Fig 3 (a) and Fig 3 (c) with respect to the transition paths.

Figures 3b and 3d were not switched. However, we have now replotted them on the same y-axis range to allow better comparison of the transition pathways and topologies.

Minor comments:

1. Line 120 – The arguments here are similar to those developed by David Castner to probe the orientation of surface bound proteins using ToF-SIMS (e.g. PMID: 15801458). It would be appropriate to add a reference.

We thank the reviewer for bringing this to our attention. Wang et al. Langmuir 2004 (PMID: 15801458) has now been accordingly referenced in the main text.

2. Line 230 – what is meant by reproducibility here? Does it mean that the presented method reproduces structure measurements obtained using other methods or is it to do with the repeatability of the measurements in this study?

We thank the reviewer for pointing this out. By “reproducibility” we mean repeatability of the measurements in this study. To avoid a possible ambiguous interpretation of our statement on Line 230, we have now reworded this statement to “The findings from this benchmarking study have evaluated the repeatability, sensitivity, and accuracy of cryo-OrbiSIMS for characterising RNA and RNA-protein complexes” in the main text. Please see line 242.

3. Line 223 – Note that the cluster only carries 1 charge and so it should be argon atoms not ions. Also, note that “argon” should be lowercase throughout.

We thank the reviewer for highlighting this. We have now made the necessary changes in the main text.

4. Line 447 = note the sample target potential will be + 57 V for positive ions and – 57 V for negative ions.

We thank the reviewer for pointing this out and Line 446 in Methods has now been modified to reflect this change.

Reviewer #2:

Overall feedback: The article of Borkar et al. is ground breaking because, by combining methods of structural biology and solid-state mass spectrometry, it manages to propose reasonable 3D structures for RNA and their complexes in the native environment. To my knowledge, it is the first time that SIMS data are directly used to establish the full 3D structure of complex biomolecules. Very cleverly, the authors use the frequency of peak assignment of the RNA residues in the cryo-orbi-SIMS spectra as base pairing restraints to obtain a better prediction of the secondary and hence the tertiary structure. Finally, they convincingly calculate the structure of 7SK RNA in its native state, apo state and remodeled by HIV Tat protein. The research is ambitious and the results introduce a step change for biological SIMS but also a very interesting evolution for structural biology. The amount of work is very impressive and the authors selected only the most salient results to show in the article, the rest being provided in SI (including a full list of identified ions, useful for researchers in the field). For these reasons, the paper deserves to be published in Nature Comm after revision.

We highly appreciate the reviewer's feedback and thank them for acknowledging the importance of this work and recommending our manuscript for publication.

“Thus, flexible, unbound, or exposed regions are expected to experience greater fragmentation”...“we proposed a hypothesis that the number of times a particular residue is assigned within different ionised fragments of the sample, and subsequently its association with multiple peaks in the cryo-OrbiSIMS spectrum, directly corresponds to its exposure or burial within the studied RNA system.” These sentences explain the working hypotheses of the authors, at the core of this study, and those might need to be somewhat refined.

1. First, it is known that 20 keV Ar3000+ clusters will create in organic and molecular solids craters that are of the order of 10nm wide and 5nm deep. Therefore, they might be able to probe the molecules/complexes deeper than what the authors envision. It's been recently shown that clusters with those energies can desorb entire proteins and clusters of proteins with masses >50 kDa (intact or fragmented).

We thank the reviewer for this insight. However, it is unlikely that any of the complexes studied in this work would be desorbed in its entirety. This is because the complexes are significantly larger than 50kDa in mass and 10 nm x 5 nm in size. For instance, the Cas9:sg RNP is the smallest complex investigated in this work, and it is already 372 kDa in mass and 10.2 nm wide along its shortest dimension.

2. Second, the “depth” probed by the beam will depend on the cluster ion dose. In Orbi-SIMS, the dose used to obtain a sufficiently intense spectrum can be large, meaning that conditions are not static and notable erosion of the surface occurs. In that case one could expect that the entire biomolecule/complex is probed, and then the inside should show as much as the outside. Unfortunately, I could not find in the method section the ion dose used for data acquisition, which would give an indication of the sputtered depth. The authors should give the ion doses in ions/cm² to help clarify that point and discuss it further if needed.

We thank the reviewer for pointing this out, we have now included the ion dose used (3.65E+13 ion/cm²) in Methods subsection **“OrbiSIMS data collection”**.

3. Third, the orientations of the molecules in the solid ice matrix are probably random with respect to the incoming beam direction, meaning that a mass spectrum acquired on a $A \times B \mu\text{m}^2$ area (the probed surface area should be mentioned in the experimental section) might represent all the possible orientations. However, maybe it does not, because molecules will orient preferentially at the air water interface when the sample is created. If that's the case, orientation could affect the detected signals, over or under-represent some parts of the molecules.

We appreciate the reviewer's comment. As we are analysing a thin layer of pure biomacromolecules adsorbed on a gold surface, the incident argon cluster beam sputters through the full sample depth during the data acquisition time (**Supplementary Figure 16**). In this case, the mass spectrum will contain information generated from the entirety of the sample layer, not just the surface. Thus, it is expected that the position and orientation of the biomacromolecules will not bias the mass information generated in the cryo-OrbiSIMS spectrum.

We have now included the probed surface area (200 x 200 μm) in Methods subsection "**OrbiSIMS data collection**".

4. Fourth, if it is clear to me that higher sample cohesive energies (larger binding forces) lead to less sputtering (in general), I'm not at ease with discussing effects of "flexibility" etc. Intuitively I would think that "more condensed" parts (more native contacts, crystallinity, etc.) of the molecules would produce smaller fragments while longer and less bound chains would result in larger fragments, as seen with more "model" samples, which is a slightly different situation.

We thank the reviewer for highlighting a plausible relationship between cohesive energies/native contacts and fragment length. However, we have observed no correlation between the number of native contacts for any RNA residue and its mean fragment length in our cryo-OrbiSIMS data. We have now provided this analysis as Supplementary Data in Spreadsheet.

Furthermore, we have removed the mention of "flexibility" from instances in the main text where the relationship between the conformational dynamics of the biomolecule and its fragmentation by the argon beam is not understood.

5. The fact that the authors find that their results contradict their hypothesis (negative correlation with SASA and positive correlation with number of native contacts) seems to indicate that the working hypothesis should be partly reformulated or at least refined. Perhaps the assignment frequency is not the right metric to correlate to the chosen parameters or the interpretation of this correlation is more complex.

We acknowledge the reviewer's suggestion and have proposed a refined hypothesis that "the peak assignment frequency for an RNA residue is related to its number of native contacts (rather than solvent exposure) in the corresponding RNA-protein complex". Please see Line 164 in the main text. A discussion centred around the interpretation of these relationships is included in the main text between Lines 157 – 164.

6. Page 7. I don't understand the PCA analysis results. PCA on SIMS spectra should give

scores and loadings corresponding to the different PCs, which can then be interpreted (or not) in terms of specific properties/features of the samples. In Figure 1, the authors directly name the axes of the principal components in the score plots as a representation of native contacts, assignment frequency, base pairing or SASA. What does it mean? In addition, the 3D plots of Figure 1 do not give any information because of the scale and the grouping of the points too close to one of the axes. The authors should show examples of the scores and loadings plots and explain how they reach their conclusions. They must explain the detailed procedure in the methods section. In general, the methods section lacks important details, both for the experiments and the data processing.

We apologise for this confusion and would like to clarify that the principal component analysis is not performed on the SIMS data. The method is instead used to understand the relationship between the following four characteristics of each RNA residue in the studied systems: (1) frequency of assignments, (2) native contacts, (3) base-pairing and (4) solvent accessible surface area (SASA). The methodology for performing this PCA are now added in Methods subsection “*Principal Component Analysis*”.

7. Page 8. Generally, the method to retrieve the 3D structure is original and quite convincing. However, since it is a new method development, it seems particularly important to verify the results of the protocol (at least in some cases) with an independent method. I’m afraid it is somewhat lacking here but I agree that it might be a challenging task.

We thank the reviewer for highlighting the confidence in our method for retrieving the 3D structures and also for acknowledging the challenges associated with verifying these results. We would like to further add that our protocol has used previously benchmarked, independent algorithms and methods (such as X-ray crystallography, SHAPE and metadynamics) for all the systems studied. The algorithms used to predict the 2D and 3D structures of RNA (with or without restraints) have been developed and validated previously through multiple studies (references 1-8 below). Similarly, the use of metadynamics simulations to accelerate the exploration of the free energy landscapes of RNAs have been validated previously (references 9-11 below).

In the current study, we have developed a correlated pipeline that uses these well-established methods to model the cryo-OrbiSIMS restrained secondary and tertiary structures of the representative RNA systems. The tertiary structures are further refined by exploring their free energy landscape using metadynamics simulations and choosing the global minimum conformation on this landscape. The refined structures are finally benchmarked against the corresponding high-resolution X-ray crystallography structures deposited in the PDB (where available for the Cas9:sg system and 5s rRNA), and against SHAPE structures for the 7SK system. This benchmarking reveals that using cryo-OrbiSIMS as restraints in the structure modelling leads to more precise prediction of the RNA structures, which is difficult to achieve without any experimental restraints. We thus strongly believe that the results of our protocol have been verified against independent, state-of-the-art methods for all the systems studied.

1. Reuter, J. S. & Mathews, D. H. RNAstructure: software for RNA secondary structure prediction and analysis. *BMC Bioinformatics* 11, 129 (2010).
2. Bellaousov, S., Reuter, J. S., Seetin, M. G. & Mathews, D. H. RNAstructure: web servers
3. Sarzynska, J., Popena, M., Antczak, M. & Szachniuk, M. RNA tertiary structure prediction using RNAComposer in CASP15. *Proteins prot.*26578 (2023) doi:10.1002/prot.26578.

4. Popena, M. et al. Automated 3D structure composition for large RNAs. *Nucleic Acids Res.* 40, e112–e112 (2012).
5. Hedaya, O. M. et al. Secondary structures that regulate mRNA translation provide insights for ASO-mediated modulation of cardiac hypertrophy. *Nat Commun* 14, 6166 (2023).
6. Romero-López, C., Roda-Herreros, M., Berzal-Herranz, B., Ramos-Lorente, S. E. & Berzal-Herranz, A. Inter- and Intramolecular RNA–RNA Interactions Modulate the Regulation of Translation Mediated by the 3' UTR in West Nile Virus. *IJMS* 24, 5337 (2023).
7. Nalewaj, M. & Szabat, M. Examples of Structural Motifs in Viral Genomes and Approaches for RNA Structure Characterization. *IJMS* 23, 15917 (2022).
8. Riesenber, S., Helmbrecht, N., Kanis, P., Maricic, T. & Pääbo, S. Improved gRNA secondary structures allow editing of target sites resistant to CRISPR-Cas9 cleavage. *Nat Commun* 13, 489 (2022).
9. Borkar, A. N. et al. Structure of a low-population binding intermediate in protein-RNA recognition. *Proc. Natl. Acad. Sci.* 113, 7171–7176 (2016).
10. Borkar, A. N., Vallurupalli, P., Camilloni, C., Kay, L. E. & Vendruscolo, M. Simultaneous NMR characterisation of multiple minima in the free energy landscape of an RNA UUCG tetraloop. *Phys. Chem. Chem. Phys.* 19, 2797–2804 (2017).
11. Bussi, G. & Laio, A. Using metadynamics to explore complex free-energy landscapes. *Nat. Rev. Phys.* 2, 200–212 (2020).
12. Mlýnský, V. & Bussi, G. Exploring RNA structure and dynamics through enhanced sampling simulations. *Curr. Opin. Struct. Biol.* 49, 63–71 (2018).

8. 242 “The exact mechanism by which the Argon cluster ion beam induces ballistic fragmentation in cryo-OrbiSIMS is unknown. It has been recently reported that the fragmentation occurs through physical bombardment and transfer of kinetic energy from the Argon ions to the analyte³⁶”. In fact, the mechanism of fragmentation induced by large Ar clusters, and its strong dependence on the energy per atom is quite well known now, because of the many experimental and computer simulation studies that have been published over the last 15 years, including the very recent study on proteins mentioned by the authors. The authors should reformulate the sentence and actually the whole paragraph. Beyond fragmentation, quite well studied, ionization remains a big question mark and the prevalent role of water, indicated by the difference between CRYO and RT SIMS, poses questions. The fact that SIMS may remain blind to certain species because they don't ionize well should not be overlooked.

We thank the reviewer for this overview regarding the fragmentation mechanism by argon GCIB. Through the statement originally on Line 242, we wanted to communicate that the susceptibility of RNA and its complexes to ballistic fragmentation by the GCIB is unknown. We have now amended Line 254 to convey this specific gap in knowledge.

Furthermore, we acknowledge that SIMS may remain blind to certain species because they don't ionise well. However, RNA is a charged molecule and for our peak assignment algorithm, we consider that all RNA fragments generated will be charged and thus visible in the cryo-OrbiSIMS spectrum.

Minor comments:

1. 81 while retaining the mass accuracy...

We have now added the missing word “accuracy” on Line 88 in the main text.

2. 86 “These results are consistent with better ionisation yields¹² in cryo-OrbiSIMS compared to room temperature conditions due to hydrogen donation from H₃O⁺ and due to molecules retaining their native state¹³”. There could also be a direct effect of the higher internal energy of the molecules at RT which adds to the energy brought by the impinging cluster. Yet the percentages of common peaks between cryo and RT data is extremely low, which certainly points to a major role of the water matrix in the desorption/ionization process.

We thank the reviewer for pointing out a major role of the water matrix. This speculation has been added to the main text Line 94.

3. 96 “These findings strongly indicate that cryo-OrbiSIMS measurements are reproducible” Can the authors comment also on the reproducibility of the RT measurements?

We believe that OrbiSIMS measurements would be reproducible under both cryogenic as well as ambient conditions.

4. 109 “However, in the negative polarity data (Fig 1b and Supplementary Figure 2), the denatured ribosome sample clustered alongside the apo sg RNAs, separate from the native ribosome sample.” Here, one could question whether the different results are due only to the denaturation or because of other differences in the initial sample. For example, are the quantities of frozen water between the denatured and native ribosome the same? Could there be an effect of the substrate (what is the sample thickness, I assume much larger than the cluster interaction volume)? Another fundamental question relates to the exact mechanism and timing of the interaction, which depends on the size and energy of the impinging cluster. For instance, do the molecules denature before breaking (unimolecular dissociation) or break immediately before desorbing (“ballistic” effect). The interpretations of the authors point to the second case, since denatured and native ribosome clearly separate. It is reasonable given that ~6 eV / Ar atom is already very energetic. The situation might be different with 1 eV/atom of energy.

We thank the reviewer for these insightful comments. The quantities of frozen water between the denatured and native ribosomes would be the same because an aliquot of the native ribosome solution was denatured by boiling at 95°C for 5 mins in a closed microcentrifuge tube. Thus, we do not expect any loss of solvent from the ribosomes between the native and denatured samples. All samples were analysed as thin layers (**Supplementary Figure 16**) adsorbed on gold surface. Since the cryo-OrbiSIMS spectrum observed for the native RNAs is distinct from the denatured RNAs, we believe that the molecules do not denature on surface but break before desorbing (ballistic effect). In case of unimolecular dissociation, due to the denaturing of the sample before desorption, the denatured and native samples would exhibit a similar mass spectrum.

5. 187 “This suggests that the structural changes between these states are energetically feasible and could represent native state dynamics of these RNAs.” I would take a closer look into the number of hydrogen bridges. Have pair distribution functions of the elements.

We thank the reviewer for this suggestion. We have assessed the feasibility of the global structural changes between two conformations based on the height of the transition state barrier between them

on the free energy landscape, rather than the pair distribution functions of the hydrogen bonded elements along the transition pathway.

6. 247 “Thus, there may be physico-chemical processes influenced by local chemical environment of the analyte that may contribute to the observed fragmentation patterns,”
Very vague sentence. Maybe not necessary or should be rephrased.

We have now removed this sentence from Line 261 in the main text.

7. Figure 3. Please show the minimum energy transition pathway in (a) the same way it is done in (c).

We have now modified Figure 3c as requested.

8. 518 “the charge was neutralized by adding the appropriate number of Mg²⁺ and Cl⁻ ions.”
The explicit amount should be mentioned in the methods section.

We have now mentioned the number of neutralising ions in Methods subsection “***Structure Modelling and Simulations***”.

Reviewer #3

Overall feedback: This work proposes Cryo-OrbiSIMS, a new mass-spec based approach to infer the number of contacts and flexibilities of each nucleotide residue in the RNA. While this information cannot be directly used for 3D structure modeling, it can be used improve RNA secondary structure prediction, which in turn improves RNA tertiary structure prediction.

1. My major concern about this work is its utility given the wide adoption of SHAPE-MaP, which, by all means, is the most commonly used experimental method to assist secondary structure determination. Both SHAPE-MaP reactivity and OrbiSIMS assignment frequencies measure the rigidity of the backbone, and they are both used (and only used) in RNA secondary structure determination. Given this fact, this work should explain whether OrbiSIMS is advantageous to SHAPE-MaP in terms of labor time and monetary cost. If not, is OrbiSIMS at least complementary to SHAPE-MaP for the task of secondary structure determination? In the introduction, include SHAPE-MaP compare it to OrbiSIMS.

We thank the reviewer for highlighting the utility of SHAPE-MaP and suggestions on benchmarking our pipeline against this technique. We have now added a comparison between SHAPE-MaP and Cryo-OrbiSIMS as Supplementary Notes and in the main text at Line 46.

2. For the case study on Cas9:sg RNP and 5s rRNA, present the secondary structure calculated using SHAPE-MaP, using OrbiSIMS, and (optionally) using a combination of SHAPE-MaP and OrbiSIMS. Calculate the similarities between secondary structure derived from SHAPE-MaP/OrbiSIMS and that assigned based on experimental 3D structure by DSSR and/or RNAViewer. This benchmark is critical to evaluate whether OrbiSIMS is more suitable for the specific task of secondary structure determination.

We thank the reviewer for this suggestion. We have extracted the 2D structures for the sg RNA and 5s rRNAs from the PDB entries 4ZT9 and 6GWT respectively using RNAPdb server and have used these 2D and 3D structures as the benchmarks for our cryo-OrbiSIMS modelling at the respective structural levels. Moreover, we have demonstrated that cryo-OrbiSIMS is at least comparable in performance to SHAPE-MaP and other secondary structure probing methods by using the case of 7SK. In the case of 7SK, there are no experimentally determined 3D structures, and thus the highest benchmark is the 2D structure determined using SHAPE and related techniques. We have provided a quantitative comparison between these 2D structures and our cryo-OrbiSIMS restrained models for 7SK in Figure 4b.

Minor comments:

1. The title of this paper starts with Cryo-OrbiSIMS but the abstract only introduces 3D-OrbiSIMS. Please explain what the difference and relation are between the two in the abstract.

We have now included the term Cryo-OrbiSIMS in the Abstract.

2. For Fig S5, in addition to current panels, it would also be helpful to add corresponding scatter plots, where each point is one residue, the x-axis is the structural feature (SASA or native contacts), and the y-axis is assignment frequency. Such a scatter plot will make the correlation (or lack hereof) more evident.

We thank the reviewer for this suggestion. We have now provided a scatter plot of SASA and native contacts v/s assignment frequency in **Supplementary Figure 5**.

3. Chimera Matchmaker is probably not the best tool to evaluate RMSD between predicted 3D structure model and native (i.e., experimental) 3D structure. This is because Matchmaker does not guarantee the correct residue level correspondence between the model and native, and residue 1 in the model can be aligned to residue 10 in the native. To address this issue, the author can use the RNA_assessment tool provided by the RNA-puzzle assessors Alternatively, they can use US-align, which has an option called “-seq” that ensures pairwise sequence correspondence.

We thank the reviewer for this suggestion. In our study, the predicted and the experimental 3D structures of the RNAs contain identical sequences. Moreover, after aligning these structures using Chimera MatchMaker, we also analysed the sequence alignment generated by the MatchMaker tool to make sure that pairwise sequence correspondence was maintained between the predicted and native structures.

6. The title of this paper “Cryo-OrbiSIMS Enables Integrative Modelling of Native Structures of RNAs at Atomic Resolution” is highly exaggerated. Cryo-OrbiSIMS cannot be used for 3D structure modeling. It can only be used for secondary structure modeling. Even if a better secondary structure can indeed improve 3D structure modeling, the authors should not claim that Cryo-OrbiSIMS “enables” modeling of atomic structure.

We acknowledge the reviewer's concern and have now modified the title of the paper to "Integrating Cryo-OrbiSIMS with Computational Modelling and Metadynamics Simulations Enhances RNA Structure Prediction at Atomic Resolution".

REVIEWER COMMENTS

Reviewer #1 (Remarks to the Author):

Thank you for the revised manuscript. The authors have addressed my concerns. The addition of supplementary note 2 is very helpful. I would suggest being more explicit / quantitative on what "stringent sample requirements" means. I found the table in Supplementary note 1 very useful and perhaps a similar table in supplementary note 2 including more quantitative guidance on e.g. the sample purity and homogeneity needed for cryo-EM / cryo-OrbiSIMS would help guide analysts. However, whilst this would be nice it is not necessary for the paper to be published in the reviewer's opinion.

Reviewer #2 (Remarks to the Author):

In general, I find the answers of the authors satisfactory following the first round of reviews and I reiterate that the findings reported in this article are impressive. In particular, the authors clarified my understanding of the samples analyzed by ToF-SIMS and the detailed experimental conditions. In that light, I have only one remaining issue, which is a rather fundamental one concerning the (cryo)OrbiSIMS characterization.

Figure S16 confirms that the analyte sample constitutes a (sub)monolayer on gold, given that an ion fluence of $1E12-1E13$ ions/cm³ is sufficient to totally consume the analyte and that gold is already visible at the very beginning of the profile. It seems reasonable since the samples were abundantly rinsed with PBS buffer, removing any weakly bound excess analyte beyond a monolayer. So I would assume that the probed biomolecules are in direct contact with the gold substrate and, in the case of cryo-OrbiSIMS measurements, complemented and surrounded by an ice matrix. It would be interesting to show the profile of the ice signal, if not through H₃O⁺ (out of the OrbiSIMS mass range), via larger mass (H₂O)_nH⁺ clusters, if those are available with sufficient intensity. In the RT measurements, the analytes are probably quite denatured by the elimination of most water and by the interaction with the gold. Increasing interaction with the gold should also lead to more fragmentation in small pieces (less large fragments) while the water matrix should preserve the native structure better and allow softer desorption of larger chain segments, an interpretation that seems to agree with the observations. In addition, water acts as a matrix enhancing ionization as was demonstrated by many publications in the past.

That brings me back to the hypothesis proposed in page 6 of the article:

"Thus, unbound, or exposed regions are expected to experience greater fragmentation, resulting in a higher number of peaks in the cryo-OrbiSIMS spectra. In contrast, regions that are bound or buried within the complexes are anticipated to exhibit fewer peaks¹⁴. Based on this understanding, we proposed a hypothesis that the number of times a particular residue is assigned within different ionised fragments of the sample, and subsequently its association with multiple peaks in the cryo-OrbiSIMS spectrum, directly corresponds to its exposure or burial within the studied RNA system."

To me this is contradictory with the statements (and the information) presented by the authors in the rebuttal:

"As we are analysing a thin layer of pure biomacromolecules adsorbed on a gold surface, the incident argon cluster beam sputters through the full sample depth during the data acquisition time (Supplementary Figure 16). In this case, the mass spectrum will contain information generated from the entirety of the sample layer, not just the surface."

Indeed, if one sputters through the whole sample, all parts of the biomolecule will be exposed at some point, unlike ToF-SIMS studies of biomolecule orientation operated in the static regime (e.g. V. Lebec et al., Probing the orientation of β -lactoglobulin on gold surfaces modified by Alkyl Thiol self-

assembled monolayers, *J. Phys. Chem. C*, 2013, 117, 11569-11577.)

I could agree with an alternative hypothesis, however, that if the RNAs are arranged with a statistical distribution of orientations on the surface, the outer regions will on average tend to interact more often with the gold substrate, and therefore be more extensively fragmented, while the inner regions, interacting with other RNA segments and water in a softer manner, could perhaps produce longer and more specific fragment chains.

I'm not sure that helps to rationalize the observations but I would like the authors to integrate what seems to me a more detailed reasoning (and more consistent with the analysis conditions) in their working hypothesis and check whether this offers a better explanation to some of the puzzling observations.

Reviewer #3 (Remarks to the Author):

The revised manuscript partly address the concerns from the previous round of peer review. However, the major concerns regarding the comparison to SHAPE-MaP are not addressed.

Major issues:

1. The revised introduction (line 44) and Supplementary Note commented that SHAPE-MaP is limited by the size of the RNA studied (<150nt). This is not true. In fact, SHAPE-MaP is shown to be able to probe secondary structure of very large RNAs such as the SARS-CoV-2 genome (~30000 nt) as shown by <https://doi.org/10.1093/nar/gkaa1053> and <https://doi.org/10.1016/j.molcel.2020.12.041>

2. I previously commented that "For the case study on Cas9:sg RNP and 5s rRNA, present the secondary structure calculated using SHAPE-MaP, using OrbiSIMS, and (optionally) using a combination of SHAPE-MaP and OrbiSIMS. Calculate the similarities between secondary structure derived from SHAPEMaP/OrbiSIMS and that assigned based on experimental 3D structure by DSSR and/or RNAViewer. This benchmark is critical to evaluate whether OrbiSIMS is more suitable for the specific task of secondary structure determination." The authors responded that "We have extracted the 2D structures for the sg RNA and 5s rRNAs from the PDB entries 4ZT9 and 6GWT respectively using RNApdbee server and have used these 2D and 3D structures as the benchmarks for our cryo-OrbiSIMS modelling at the respective structural levels." I did not see how this address my concerns. For the case of Cas9:sg RNA and 5s rRNA, there should be (A) secondary structure assigned from PDB 4ZT9 and 6GWT by RNApdbee; (B) secondary structure derived from cryo-OrbiSIMS; (C) secondary structure derived from SHAPE-MaP; (D) secondary structure predicted by RNAstructure. The similarity (in terms of F1-score, accuracy, MCC or Jaccard Index) between A and B, between A and B, between A and C and between A and D should all be calculated. The current manuscript presents results from A, B and D, but not C. This does not show to what extent does Cryo-OrbiSIMS improve over SHAPE-MaP.

3. I previously commented that "3. Chimera Matchmaker is probably not the best tool to evaluate RMSD between predicted 3D structure model and native (i.e., experimental) 3D structure. This is because Matchmaker does not guarantee the correct residue level correspondence between the model and native, and residue 1 in the model can be aligned to residue 10 in the native. To address this issue, the author can use the RNA_assessment tool provided by the RNA-puzzle assessors Alternatively, they can use US-align, which has an option called "-seq" that ensures pairwise sequence correspondence." The authors responded that "Moreover, after aligning these structures using Chimera MatchMaker, we also analysed the sequence alignment generated by the MatchMaker tool to make sure that pairwise sequence correspondence was maintained between the predicted and native structures." I cannot make sure this is true unless the authors show the pairwise alignment from Chimera Matchmaker in the supplement.

4. As an extension to the previous comment, in Figure 2, the RMSD values are highly misleading. For example, in figure 2bd, the RMSD is the local/pair RMSD from Chimera MatchMaker. Although de novo model has a higher local/pair RMSD, it actually can fit more residues by MatchMaker than the restrained model. Therefore, it is not possible to determine whether the de novo model or the restrained model is better. In fact, based on global RMSD, it seems the unrestrained model is better at least for 4zt9. In any case, only the structural similarity metric (RMSD, GDT or TM-score) corresponds to all residues (not just the small number of fitted residues) is meaningful. This is why I recommend using "USalign -seq".

Minor issues:

1. In Figure 2ac, what are the percentage values? This should be explained in the main text.

Response to Reviewer Comments

We thank all three reviewers for their continued feedback on our manuscript. We appreciate the positive reception of our revised work, with Reviewer 1 finding the supplementary notes helpful and Reviewer 2 commending the clarified information on the samples and experimental conditions. However, we acknowledge Reviewer 3's concern that our previous revisions only partially addressed their feedback, particularly regarding the comparison to SHAPE-MaP. In response to this feedback, we have performed further revisions to the manuscript and have fully addressed the issues raised by all three reviewers. Please see our point-by-point response below for a comprehensive explanation of the changes we have made.

Reviewer #1

1. Thank you for the revised manuscript. The authors have addressed my concerns. The addition of supplementary note 2 is very helpful. I would suggest being more explicit / quantitative on what "stringent sample requirements" means. I found the table in Supplementary note 1 very useful and perhaps a similar table in supplementary note 2 including more quantitative guidance on e.g. the sample purity and homogeneity needed for cryo-EM / cryo-OrbiSIMS would help guide analysts. However, whilst this would be nice it is not necessary for the paper to be published in the reviewer's opinion.

We appreciate the reviewer's positive feedback on the revised manuscript and find their suggestion regarding Supplementary Note 2 valuable. We have now provided more guidance on sample requirements for cryo-EM analysis in **Supplementary Note 2**.

Reviewer #2

1. In general, I find the answers of the authors satisfactory following the first round of reviews and I reiterate that the findings reported in this article are impressive. In particular, the authors clarified my understanding of the samples analyzed by ToF-SIMS and the detailed experimental conditions.

We thank the reviewer for their positive assessment of our work and are glad that the previous revision helped to clarify the samples analysed and the OrbiSIMS experiments conducted in this work.

2. In that light, I have only one remaining issue, which is a rather fundamental one concerning the (cryo)OrbiSIMS characterization. Figure S16 confirms that the analyte sample constitutes a (sub)monolayer on gold, given that an ion fluence of $1E12-1E13$ ions/cm³ is sufficient to totally consume the analyte and that gold is already visible at the very beginning of the profile. It seems reasonable since the samples were abundantly rinsed with PBS buffer, removing any weakly bound excess analyte beyond a monolayer. So, I would assume that the probed biomolecules are in direct contact with the gold substrate and, in the case of cryo-OrbiSIMS measurements, complemented and surrounded by an ice matrix. It would be interesting to show the profile of the ice signal, if not through H_3O^+ (out of the OrbiSIMS mass range), via larger mass $(H_2O)_nH^+$ clusters, if those are available with sufficient intensity. In the RT measurements, the analytes are probably quite denatured by the elimination of most water and by the interaction with the gold. Increasing interaction with the gold should also lead to more fragmentation in small pieces (less large fragments) while the water matrix should preserve the native structure better and allow softer desorption of larger chain segments, an interpretation that seems to agree with the observations. In addition, water acts as a matrix enhancing ionization as was demonstrated by many publications in the past.

We appreciate the reviewer's insightful comments on Supplementary Figure 16 and the proposed interpretation of the data. However, larger clusters of water $(H_2O)_{nH^+}$ are not detectable in the OrbiSIMS data, as the analyser and the RF transfer to the Orbitrap are optimised for detection of organic species. The $(H_2O)_{nH^+}$ can technically be detected when the time of flight analyser is used, whether by the use of GCIB and ToF analyser or LMIG and ToF analyser, however these experiments were not conducted here.

3. That brings me back to the hypothesis proposed in page 6 of the article: "Thus, unbound, or exposed regions are expected to experience greater fragmentation, resulting in a higher number of peaks in the cryo-OrbiSIMS spectra. In contrast, regions that are bound or buried within the complexes are anticipated to exhibit fewer peaks¹⁴. Based on this understanding, we proposed a hypothesis that the number of times a particular residue is assigned within different ionised fragments of the sample, and subsequently its association with multiple peaks in the cryo-OrbiSIMS spectrum, directly corresponds to its exposure or burial within the studied RNA system." To me this is contradictory with the statements (and the information) presented by the authors in the rebuttal: "As we are analysing a thin layer of pure biomacromolecules adsorbed on a gold surface, the incident argon cluster beam sputters through the full sample depth during the data acquisition time (Supplementary Figure 16). In this case, the mass spectrum will

contain information generated from the entirety of the sample layer, not just the surface.” Indeed, if one sputters through the whole sample, all parts of the biomolecule will be exposed at some point, unlike ToF-SIMS studies of biomolecule orientation operated in the static regime (e.g. V. Lebec et al., Probing the orientation of β -lactoglobulin on gold surfaces modified by Alkyl Thiol self-assembled monolayers, *J. Phys. Chem. C*, 2013, 117, 11569-11577.) I could agree with an alternative hypothesis, however, that if the RNAs are arranged with a statistical distribution of orientations on the surface, the outer regions will on average tend to interact more often with the gold substrate, and therefore be more extensively fragmented, while the inner regions, interacting with other RNA segments and water in a softer manner, could perhaps produce longer and more specific fragment chains.

I’m not sure that helps to rationalize the observations but I would like the authors to integrate what seems to me a more detailed reasoning (and more consistent with the analysis conditions) in their working hypothesis and check whether this offers a better explanation to some of the puzzling observations.

We thank the reviewer for this insightful analysis and identification of potential contradictions between our initial hypothesis and the full-depth sample sputtering revealed in **Supplementary Figure 16**. To rationalise our experimental observations and to provide a more comprehensive interpretation of the cryo-OrbiSIMS data, we have now presented four possible scenarios in **Supplementary Figure 17**.

Scenario A represents our initial hypothesis, where the exposed regions are expected to be bombarded more and thus fragmented more, leading to a higher number of residue peak assignments from these regions. However, our peak assignment frequencies are not correlated to the residues’ solvent accessible surface area, thus this scenario is unlikely.

Reviewer 2 thus proposed Scenario B, where the RNA systems have a statistical distribution of orientations on the surface. In this scenario, the outer regions of the RNA will experience intensive fragmentation, producing shorter fragments. However, we do not see any correlation between the average fragment length observed for an RNA residue and its solvent exposed surface area (**Supplementary Data in Spreadsheet**), suggesting that this scenario is also unlikely, and the RNA complexes might not have been preferentially oriented on the surface.

The reviewer’s suggestion that inner regions might produce more specific fragment chains due to softer interactions is, however, intriguing. In our data, we see a higher representation of fragments arising from the RNA:RNA and RNA:protein interaction sites (**Scenario D**). This aligns with previous studies which demonstrated higher secondary ion yields from regions with higher cohesive energies^{1,2}. Furthermore, the RNA:RNA and RNA:Protein interactions would also stabilise the labile phosphodiester backbone³ of the RNA, likely producing specific fragments from these regions in the detectable mass range of the OrbiSIMS. Based on this understanding, we speculate that **Scenario D** is a likely case and propose it as our refined working hypothesis in the main text (**Lines 258 – 283**).

1. Cristaudo, V. *et al.* Ion yield enhancement at the organic/inorganic interface in SIMS analysis using Ar-GCIB. *Appl. Surf. Sci.* **536**, 147716 (2021).
2. Bhattarai, G. *et al.* Underlying role of mechanical rigidity and topological constraints in physical sputtering and reactive ion etching of amorphous materials. *Phys. Rev. Mater.* **2**, 055602 (2018).
3. Corley, M., Burns, M. C. & Yeo, G. W. How RNA-Binding Proteins Interact with RNA: Molecules and Mechanisms. *Mol. Cell* **78**, 9–29 (2020).

Reviewer #3:

The revised manuscript partly addresses the concerns from the previous round of peer review. However, the major concerns regarding the comparison to SHAPE-MaP are not addressed.

We acknowledge Reviewer 3's concern that our previous revisions only partially addressed their concerns, particularly regarding the comparison to SHAPE-MaP. Please see our point-by-point response below for our second round of revisions to fully address the issues raised.

Major issues:

1. The revised introduction (line 44) and Supplementary Note commented that SHAPE-MaP is limited by the size of the RNA studied (<150nt). This is not true. In fact, SHAPE-MaP is shown to be able to probe secondary structure of very large RNAs such as the SARS-CoV-2 genome (~30000 nt) as shown by

<https://doi.org/10.1093/nar/qkaa1053> and <https://doi.org/10.1016/j.molcel.2020.12.041>

We appreciate the reviewer's attention to detail and thank them for highlighting a potential misinterpretation for our statement. We intended to convey the LOWER limit, and not the UPPER limit, of the size of the RNAs that can be reliably studied by SHAPE-MaP given the limitations imposed by RT PCR amplification and sequencing steps. We agree with the reviewer that SHAPE-MaP workflows are typically suitable for a wide range of RNA lengths. We have now revised our statement in main text Introduction at **Line 42** and in **Supplementary Note 1**.

2. I previously commented that "For the case study on Cas9:sg RNP and 5s rRNA, present the secondary structure calculated using SHAPE-MaP, using OrbiSIMS, and (optionally) using a combination of SHAPE-MaP and OrbiSIMS. Calculate the similarities between secondary structure derived from SHAPeMaP/OrbiSIMS and that assigned based on experimental 3D structure by DSSR and/or RNAViewer. This benchmark is critical to evaluate whether OrbiSIMS is more suitable for the specific task of secondary structure determination." The authors responded that "We have extracted the 2D structures for the sg RNA and 5s rRNAs from the PDB entries 4ZT9 and 6GWT respectively using RNApdbee server and have used these 2D and 3D structures as the benchmarks for our cryo-OrbiSIMS modelling at the respective structural levels." I did not see how this address my concerns. For the case of Cas9:sg RNA and 5s rRNA, there should be (A) secondary structure assigned from PDB 4ZT9 and 6GWT by RNApdbee; (B) secondary structure derived from cryo-OrbiSIMS; (C) secondary structure derived from SHAPE-MaP; (D) secondary structure predicted by RNAstructure. The similarity (in terms of F1-score, accuracy, MCC or Jaccard Index) between A and B, between A and C and between A and D should all be calculated. The current manuscript presents results from A, B and D, but not C. This does not show to what extent does Cryo-OrbiSIMS improve over SHAPE-MaP.

We appreciate the reviewer's detailed feedback and their request for a more comprehensive comparison between SHAPE-MaP and cryo-OrbiSIMS. We would like to clarify that our aim for this manuscript was to demonstrate a novel application for cryo-OrbiSIMS and showcase that the mass

information extracted from this technique can be integrated as restraints in computational algorithms, rather than directly compare its performance with other techniques.

While we agree that comparing with PDB structures provides valuable validation, we understand the desire for a broader analysis with SHAPE-MaP data. Integrating this information could offer complementary insights. However, we encountered challenges in acquiring this data, especially for the sg RNA construct, due to lack of in-house expertise in the area and unsuccessful attempts to outsource the work to commercial service providers such as Illumina, Novogene, DeepSeq (our in-house sequencing team) and EclipseBio. EclipseBio specifically declined the project stating that the experiments would require the development of custom protocols, which would not be possible with their current bandwidth.

We were instead able to source chemical probing data for the 5s rRNA from RMDB and RASP databases. Comparison of the F1 scores as requested by the reviewer (see table below) illustrates that the cryo-OrbiSIMS 2D structures perform equally well as the chemically probed structures. Furthermore, an additional comparison between cryo-OrbiSIMS restrained 2D structures of 5s rRNA and 7SK RNA with their respective chemically probed structure, indicate strong complementary between the structural readout of the two techniques.

System	PDB with cryo-OrbiSIMS (A and B)	PDB with SHAPE-MaP (A and C)	PDB with RNAstructure (A and D)	Cryo-OrbiSIMS with SHAPE-MaP (B and C)
Cas9-sg RNP complex	0.7	-	0.63	-
5s rRNA in bacterial ribosomal complex	0.825	0.71* 0.84**	0.38	0.69* 0.98**
7SK in native RNP	-	-	-	0.82 ⁴

*DMS probing^{5,6} (*in vivo*)

**IM7 probing [5SRRNA_IM7_0006⁷, *in vivo*. where the chemical probing was performed on extracted RNA (e.g. 5SRRNA_IM7_0009^{7,8}), the F1-score was 0.34. However, this latter experimental condition is not equivalent to our cryo-OrbiSIMS samples conditions, nor to the PDB structure, and thus the value was omitted from the comparison table above]

- Olson, S. W. *et al.* Discovery of a large-scale, cell-state-responsive allosteric switch in the 7SK RNA using DANCE-MaP. *Mol. Cell* **82**, 1708-1723.e10 (2022).
- Burkhardt, D. H. *et al.* Operon mRNAs are organized into ORF-centric structures that predict translation efficiency. *eLife* **6**, e22037 (2017).
- Li, P., Zhou, X., Xu, K. & Zhang, Q. C. RASP: an atlas of transcriptome-wide RNA secondary structure probing data. *Nucleic Acids Res.* **49**, D183–D191 (2021).
- Yesselman, J. D. *et al.* Updates to the RNA mapping database (RMDB), version 2. *Nucleic Acids Res.* **46**, D375–D379 (2018).

8. Watters, K. E., Yu, A. M., Strobel, E. J., Settle, A. H. & Lucks, J. B. Characterizing RNA structures in vitro and in vivo with selective 2'-hydroxyl acylation analyzed by primer extension sequencing (SHAPE-Seq). *Methods* **103**, 34–48 (2016).

3. I previously commented that "3. Chimera Matchmaker is probably not the best tool to evaluate RMSD between predicted 3D structure model and native (i.e., experimental) 3D structure. This is because Matchmaker does not guarantee the correct residue level correspondence between the model and native, and residue 1 in the model can be aligned to residue 10 in the native. To address this issue, the author can use the RNA_assessment tool provided by the RNA-puzzle assessors Alternatively, they can use US-align, which has an option called "-seq" that ensures pairwise sequence correspondence." The authors responded that "Moreover, after aligning these structures using Chimera MatchMaker, we also analysed the sequence alignment generated by the MatchMaker tool to make sure that pairwise sequence correspondence was maintained between the predicted and native structures." I cannot make sure this is true unless the authors show the pairwise alignment from Chimera Matchmaker in the supplement.

We appreciate the reviewer's prudent remark and have included the pairwise sequence alignments created by MatchMaker tool for the sg RNA and 5r RNA (**Response Figures 1-2**) below. Further heeding the reviewer's advice, we have used the TM-Scores as a more reliable metric of the 3D structure comparison (see response to point 4 below) and have revised the values in **Figure 2, Supplementary Table 2 and Lines 178 – 182 in the main text.**

Response Figure 1: Pairwise sequence alignment viewed in Chimera MatchMaker for the sg RNA sequence in the PDB 4Z9T structure and cryo-OrbiSIMS structure.

The University of Nottingham Collection of Public Research Data

Response Figure 2: Pairwise sequence alignment viewed in Chimera MatchMaker for the 5s rRNA sequence in the PDB 6GWT structure and cryo-OrbiSIMS structure

4. As an extension to the previous comment, in Figure 2, the RMSD values are highly misleading. For example, in figure 2bd, the RMSD is the local/pair RMSD from Chimera MatchMaker. Although de novo model has a higher local/pair RMSD, it actually can fit more residues by MatchMaker than the restrained model. Therefore, it is not possible to determine

whether the de novo model or the restrained model is better. In fact, based on global RMSD, it seems the unrestrained model is better at least for 4zt9. In any case, only the structural similarity metric (RMSD, GDT or TM-score) corresponds to all residues (not just the small number of fitted residues) is meaningful. This is why I recommend using "USalign -seq".

We thank the reviewer for this insight and for recommending the use of global structure similarity scores as a more accurate metric for comparisons between our predicted models and the PDB structures. We have addressed this feedback by calculating the TM-Score values and have revised **Figure 2, Supplementary Table 2, and Lines 178 – 182 and 502 – 504** in the main text to reflect the changes.

Our analysis reveals that while the RMSD values between the PDB structure and cryo-OrbiSIMS predicted structures remain comparable to the unrestrained predictions, the TM-Score shows a clear improvement. This improvement signifies a higher level of structural agreement between the cryo-OrbiSIMS restrained structure and the PDB reference.

Minor issues:

1. In Figure 2ac, what are the percentage values? This should be explained in the main text.

The percentage values are percent structure similarity and percent structure identity between the predicted 2D structures and PDB-derived 2D structures. The metrics are calculated using the Beagle (BEar Alignment Global and Local) algorithm⁹. This information is now included in **Figure 2** legend and in the main text at **Line 500** .

9. Mattei, E., Pietrosanto, M., Ferrè, F. & Helmer-Citterich, M. Web-Beagle: a web server for the alignment of RNA secondary structures. *Nucleic Acids Res.* **43**, W493–W497 (2015).

REVIEWER COMMENTS

Reviewer #2 (Remarks to the Author):

The authors have addressed all my concerns.

Reviewer #3 (Remarks to the Author):

I still have two major concerns over the data for comparison of cryo-OrbiSIMS to other methods newly presented by the revised manuscript.

1. I previously commented that "For the case of Cas9:sg RNA and 5s rRNA, there should be (A) secondary structure assigned from PDB 4ZT9 and 6GWT by RNApdbee; (B) secondary structure derived from cryo-OrbiSIMS; (C) secondary structure derived from SHAPE-MaP; (D) secondary structure predicted by RNAstructure. The similarity (in terms of F1-score, accuracy, MCC or Jaccard Index) between A and B, between A and C and between A and D should all be calculated. The current manuscript presents results from A, B and D, but not C. This does not show to what extent does Cryo-OrbiSIMS improve over SHAPE-MaP." The authors are able to show a table for such data for 5s rRNA in the response letter, but the table cannot be found in the main text or supplement.

2. In Supplementary Table 2, it is unclear what is the difference between column A and C and between column B and D. The main text states that all RNA structure prediction are performed by RNacomposer, so it can be assumed that all columns are comparison of native RNA 3D structure and RNacomposer models built with different kind of restraints. Yet, this is not explained explicitly. Moreover, the table caption claimed that "cryoOrbiSIMS structure (column B) are not significantly improved compared to the unrestrained structure prediction (column A), which could be possibly attributed to the overall inefficiency of 3D structure prediction algorithm (column C)." This cannot be more incorrect, as column C has a clearly far better 5s rRNA structure model than other columns.

Reviewer #3 (Remarks on code availability):

I am not able to install and run the code because it depends on MATLAB, a proprietary software I have no access to. Nonetheless, the code is not well packaged.

1. technical_process.m this takes as input the gold reference peaks (gold.mat), the peak lists exported from SurfaceLab7 software and a simple nameslist file that lists the dataset names to be processed.

There is no explanation of the format of gold.mat, how it was generated, and where we can find an example. Moreover, this file is called 'peaklists.reference/gold.mat' in the code but just 'gold.mat' by the readme file. This script also calls a file named './outputs/2reps_all.mat', whose purpose and format is not clearly documented.

2. Please make sure the code can also be run on Octave, as not researcher can buy the expensive MATLAB package.

Response to Reviewer Comments

We thank the two reviewers for their continued feedback on our manuscript. We appreciate the positive reception of our revised work by Reviewer 2 and acknowledge Reviewer 3's concerns regarding the newly presented data in the revised manuscript and the software availability. In response to this feedback, we have performed further revisions to the manuscript to fully address these issues. Please see our point-by-point response below for a comprehensive explanation of the changes we have made.

Reviewer #2

1. The authors have addressed all my concerns.

We are glad that the reviewer found our analysis of the OrbiSIMS fragmentation pattern and presentation of Supplementary Figure 17 satisfactory and appreciate their positive feedback on the revised manuscript.

Reviewer #3

1. I previously commented that "For the case of Cas9:sg RNA and 5s rRNA, there should be (A) secondary structure assigned from PDB 4ZT9 and 6GWT by RNApdbee; (B) secondary structure derived from cryo-OrbiSIMS; (C) secondary structure derived from SHAPE-MaP; (D) secondary structure predicted by RNAstructure. The similarity (in terms of F1-score, accuracy, MCC or Jaccard Index) between A and B, between A and C and between A and D should all be calculated. The current manuscript presents results from A, B and D, but not C. This does not show to what extent does Cryo-OrbiSIMS improve over SHAPE-MaP." The authors are able to show a table for such data for 5s rRNA in the response letter, but the table cannot be found in the main text or supplement.

We thank the reviewer for this suggestion. We have now included the comparison table as **Supplementary Table 2** in the supplementary information document and a brief description of the comparison results between lines 176 – 179 and 223 – 224 in the main text.

2. In **Supplementary Table 2**, it is unclear what is the difference between column A and C and between column B and D. The main text states that all RNA structure prediction are performed by RNAcomposer, so it can be assumed that all columns are comparison of native RNA 3D structure and RNAcomposer models built with different kind of restraints. Yet, this is not explained explicitly. Moreover, the table caption claimed that "cryoOrbiSIMS structure (column B) are not significantly improved compared to the unrestrained structure prediction (column A), which could be possibly attributed to the overall inefficiency of 3D structure prediction algorithm (column C)." This cannot be more incorrect, as column C has a clearly far better 5s rRNA structure model than other columns.

We thank the reviewer for their feedback on **Supplementary Table 2** (now **Supplementary Table 3**). We have added a detailed description of the structures compared and an updated interpretation of the TM-scores within the revised table caption for **Supplementary Table 3**.

Reviewer #3 (Remarks on code availability):

I am not able to install and run the code because it depends on MATLAB, a proprietary software I have no access to. Nonetheless, the code is not well packaged.

1. technical_process.m this takes as input the gold reference peaks (gold.mat), the peak lists exported from SurfaceLab7 software and a simple nameslist file that lists the dataset names to be processed. There is no explanation of the format of gold.mat, how it was generated, and where we can find an example. Moreover, this file is called 'peaklists.reference/gold.mat' in the code but just 'gold.mat' by the readme file. This script also calls a file named './outputs/2reps_all.mat', whose purpose and format is not clearly documented.

We thank the reviewer for highlighting the need for greater clarity and accessibility regarding the code and data. To ensure reproducibility and transparency, we have now provided all the codes within GitHub repository https://github.com/BorkarLab/OrbiSIMS_RNA_analysis.git. A comprehensive code documentation that enlists the system requirements, installation instructions, step-by-step instructions for running the scripts and explanations of file formats and functionalities is also included. We have also provided a sample dataset for users to test and understand the code's functionality. Furthermore, the raw OrbiSIMS data generated for the RNA systems investigated in this work is publicly accessible via The University of Nottingham Collection of Public Research Data with DOI:10.17639/nott.7354.

These changes are highlighted in the updated Code Availability statement in the main text between lines 532 – 537.

2. Please make sure the code can also be run on Octave, as not researcher can buy the expensive MATLAB package.

We thank the reviewer for highlighting the importance of code accessibility and acknowledge their concern about running the code on Octave, an open-source alternative to MATLAB.

There is a wide precedent for the use of MATLAB within the context of our research^{1,2} and within the wider structural biology and biophysics community (example list of relevant publications within the past five years in Nature Communications journal). However, we acknowledge that the licencing requirements may limit accessibility for some users. Thus, to promote broader accessibility, we have released the codes under an open-source license, which would allow researchers to potentially modify the code for their preferred environment, such as Octave. Within each code, we have also made remarks about any possible cross-platform incompatibility and modifications required to function across different environments.

Furthermore, to enhance accessibility and collaboration within the research community, we will offer access to all licensed software, including SurfaceLab7 and MATLAB, required to generate, process and analyse the OrbiSIMS data through our instrument access provision for external users of OrbiSIMS. This is also in line with the standard practice in the field for accessing specialist instrument and software.

1. Edney, M. K. *et al.* Molecular Formula Prediction for Chemical Filtering of 3D OrbiSIMS Datasets. *Anal. Chem.* **94**, 4703–4711 (2022).
2. Kotowska, A. M. *et al.* Protein identification by 3D OrbiSIMS to facilitate in situ imaging and depth profiling. *Nat. Commun.* **11**, 5832 (2020).